# CTIArena: Benchmarking LLM Knowledge and Reasoning across Heterogeneous Cyber Threat Intelligence

## Abstract

Cyber threat intelligence (CTI) is central to modern cybersecurity, providing critical insights for detecting and mitigating evolving threats. With the natural language understanding and reasoning capabilities of large language models (LLMs), there is increasing interest in applying them to CTI, which calls for benchmarks that can rigorously evaluate their performance. Several early efforts have studied LLMs on some CTI tasks but remain limited: (i) they adopt only *closed-book settings*, relying on parametric knowledge without leveraging CTI knowledge bases; (ii) they cover only a narrow set of tasks, lacking a systematic view of the CTI landscape; and (iii) they restrict evaluation to *single-source analysis*, unlike realistic scenarios that require reasoning across multiple sources. To fill these gaps, we present CTIArena, the first benchmark for evaluating LLM performance on *heterogeneous*, *multi-source* CTI under *knowledge-augmented* settings. CTIArena spans three categories, structured, unstructured, and hybrid, further divided into nine tasks that capture the breadth of CTI analysis in modern security operations. We evaluate ten widely used LLMs and find that most struggle in closed-book setups but show noticeable gains when augmented with security-specific knowledge through our designed retrieval-augmented techniques. These findings highlight the limitations of general-purpose LLMs and the need for domain-tailored techniques to fully unlock their potential for CTI.

## 1 Introduction

The evolving cyberspace landscape has led to unprecedented growth in cyber attacks, posing significant challenges for organizations worldwide. Cyber threat intelligence (CTI) plays a central role in enabling timely defense against such threats and is widely used in both detection and response operations. Open-source CTI refers to publicly available intelligence shared by security vendors and communities to inform practitioners of existing and emerging threats. A defining characteristic of CTI is its heterogeneous and multi-source knowledge structure. Common open-source CTI resources include authoritative taxonomies such as CVE (Program, 2025), CWE (MITRE, 2025b), CAPEC (MITRE, 2025a), and MITRE ATT&CK (MITRE, 2025c), which catalog vulnerabilities, weaknesses, and adversary behaviors in structured formats, as well as vendor reports and blogs that document ongoing threat events with rich contextual detail. Collectively, these sources provide a rich yet fragmented knowledge base, whose heterogeneity and dispersion present major challenges for knowledge management, integration, and application. With the rapid advancement of large language models (LLMs), there is growing interest in their potential to synthesize, interpret, and reason over this complex intelligence landscape. This motivates the central question of this work: *To what extent can LLMs effectively reason over heterogeneous, multi-source CTI, and does incorporating CTI-specific knowledge augmentation improve their performance?*

A few preliminary efforts have explored the capacity of LLMs for CTI reasoning, and these remain very limited in scope and setup. CTIBench (Alam et al., 2025), published at NeurIPS 2024, covers four tasks (root cause mapping, vulnerability severity prediction, attack technique extraction, and threat actor attribution) under a closed-book setting where models rely solely on parametric knowledge. SEvenLLM (Ji et al., 2024) introduces a bilingual instruction corpus and benchmark tailored to incident analysis, but is restricted to small instruction-tuned models on unstructured CTI such as ven-

**Table I:** Comparison of task coverage between CTIARENA and existing major CTI benchmarks, CTIBench (Alam et al., 2025) and SEvenLLM (Ji et al., 2024).

| Task | Example Question | Example Answer | CTIBench | SevenLLM | Ours |
|------|------------------|----------------|----------|----------|------|
| **CTI-RCM:** Root Cause Mapping | *Which CWE is the root cause of CVE-2025-52988?* | *CWE-78 (Improper neutralization of special elements in OS command) is the root cause of CVE-2025-52988, which allows unsanitized CLI arguments to enable OS command injection.* | ✓ | ✗ | ✓ |
| **CTI-WIM:** Weakness Instantiation Mapping | *Which CVE instantiates CWE-192 (Integer Coercion Error)?* | *CVE-2022-2639 demonstrates an integer coercion error via incorrect size reservation, instantiating CWE-192.* | ✗ | ✗ | ✓ |
| **CTI-ATD:** Attack Technique Derivation | *Which MITRE ATT&CK technique maps to CAPEC-25 (Forced Deadlock)?* | *T1499.004 corresponds to exploitation of deadlock causing denial of service, mapping CAPEC-25.* | ✓ | ✗ | ✓ |
| **CTI-ESD:** Exploitation Surface Discovery | *Which CAPEC attack pattern exploits the Absolute Path Traversal weakness (CWE-36)?* | *CAPEC-597 exploits use of file system absolute paths to access unauthorized files, targeting CWE-36.* | ✗ | ✗ | ✓ |
| **CTI-MLA:** Malware Lineage Analysis | *I recently read a blog on the BlackCat Malware. What are all the distinct malware variant names mentioned across existing threat intelligence?* | *- BlackCat*
*- Munchkin*
*- ExMatter*
*- COBEACON* | ✗ | ✓ | ✓ |
| **CTI-TAP:** Threat Actor Profiling | *I recently read a blog on the RansomHub Threat Actor. What is the canonical threat actor name, resolving any aliases mentioned across existing threat intelligence?* | *- Canonical Threat Actor Name: RansomHub*
*- Known Alias: Greenbottle* | ✓ | ✓ | ✓ |
| **CTI-CSC:** Campaign Storyline Construction | *I recently read a blog on the Silent Skimming Campaign. What are the primary target industries or regions mentioned across existing threat intelligence?* | *- Regions: Asia-Pacific (APAC), North America, Latin America*
*- Industries: e-commerce platforms, online payment and point-of-sale (PoS) servers* | ✗ | ✗ | ✓ |
| **CTI-VCA:** Vulnerability Catalog Attribution | *I recently read in a security blog: "The ransomware's effectiveness is partly due to the poor isolation and inadequate control of privileges on victim systems. ... Please identify the CWE category that maps to this vulnerability."* | *- CWE-284: Improper Access Control.*
*- The described ransomware exploits inadequate privilege and process permission controls, allowing attackers to terminate security services and move laterally, which directly aligns with CWE-284's definition.* | ✗ | ✗ | ✓ |
| **CTI-ATA:** Attack Technique Attribution | *As I recently read in a security blog: "The tool allows the threat actors to extract emails from Yahoo!, Google, and Microsoft Outlook... Which MITRE ATT&CK technique category does this behavior map to?"* | *- T1114.001 (Local Email Collection).*
*- Hyperspace accesses and extracts email messages stored locally, aligning with T1114.001's definition* | ✗ | ✗ | ✓ |

dor reports and blogs. Both benchmarks exhibit three major limitations. First, **narrow task scope:** CTIBench covers only four isolated tasks, while SEvenLLM is confined to unstructured reports, leaving other CTI tasks and sources unexplored. Second, **limited evaluation setup:** CTIBench tests only five models in prompting-only mode, and SEvenLLM considers only small instruction-tuned models, excluding frontier LLMs. Third, **lack of realism:** both adopt closed-book configurations, without leveraging existing CTI knowledge bases via retrieval-augmented methods, and both restrict inputs to a single source, whereas real-world threat analysis in security operation centers (SOCs) typically requires correlating heterogeneous evidence across reports and taxonomies (ThreatConnect, Inc., 2018). CTIBench is also limited in scale, with only about 150 manually annotated queries, making it infeasible to extend to emerging threats.

These limitations motivate our design of CTIARENA, which offers broader task coverage, realistic evaluation setups, and scalable benchmark construction. CTIARENA is a principled benchmark suite for evaluating LLMs on heterogeneous, multi-source CTI. Unlike prior benchmarks that target a few isolated tasks with prompting-only setups, CTIARENA systematically maps the CTI analysis landscape into nine representative tasks across three categories, reflecting major forms of CTI that security analysts must integrate in practice. As shown in Table I, **Structured tasks** involve reasoning over authoritative taxonomies such as CVE (Program, 2025), CWE (MITRE, 2025b), CAPEC (MITRE, 2025a), and MITRE ATT&CK (MITRE, 2025c), which catalog vulnerabilities, weaknesses, and adversary techniques in structured formats. **Unstructured tasks** draw on vendor

reports and blogs, where analysts must interpret narrative descriptions of adversaries, campaigns, and malware families. **Hybrid tasks** link structured enumerations with unstructured narratives, reflecting practical threat analysis workflows where enumerations are used to attribute actors, analyze malware lineage, and validate campaign activity. Together, these nine tasks are principled in design to cover the core workflows that threat analysts routinely perform in SOCs, such as mapping vulnerabilities to weaknesses, attributing adversary behavior to ATT&CK techniques, profiling actors, tracing malware evolution, and correlating multi-report campaigns.

Table I summarizes the nine tasks with example question–answer (QA) pairs (more details in Section 3.3). The benchmark was constructed through a three-stage pipeline consisting of factually-grounded QA generation via carefully designed task-specific LLM prompt templates, LLM judge filtering to remove low-quality samples, and human expert cross-verification to ensure the final quality (see Section 3.2 for details). Each entry includes a natural-language question, a gold-standard answer, and supporting evidence drawn from either structured repositories or unstructured reports. The final benchmark comprises 691 QA pairs: 371 structured, 150 unstructured, and 170 hybrid.

We evaluate ten representative LLMs, including four open-source and six proprietary systems, under both closed-book and knowledge-augmented settings. Beyond two generic baselines, inference-time knowledge injection and vanilla retrieval-augmented generation (RAG), we introduce two *security-specific retrieval-augmentation techniques* explicitly tailored to the structure and semantics of CTI knowledge, going beyond generic semantic similarity. The first, *CSKG-guided RAG*, leverages a curated Cyber Security Knowledge Graph to retrieve evidence based on entity-level overlaps (e.g., shared actors, malware, or ATT&CK techniques). The second, *RAG with attack-behavior decomposition*, reformulates narrative inputs into fine-grained behaviors (tactics, techniques, and affected components) aligned with security taxonomies, thereby closing the gap between varied phrasings in reports and standardized terminology in CTI frameworks. Our results show that closed-book inference is severely limited. While generic knowledge augmentation brings substantial gains, security-tailored retrieval further improves performance, particularly on hybrid and unstructured tasks where vanilla semantic search struggles to align free text with formal CTI domain taxonomies. We also analyzed the failure modes of these baselines, revealing how inappropriate evidence can mislead models and pointing to directions for more robust augmentation (see Sections 4.2 and 4.3). Together, these findings highlight that scaling model size alone is insufficient; systematic progress in CTI requires knowledge-augmented LLMs equipped with domain-specific retrieval strategies.

## 2 RELATED WORK

**LLM for CTI Analysis.** LLMs have seen growing application in CTI analysis. Early work focused on extracting structured information from narrative reports, such as tactics, techniques, and procedures (TTP) (Cuong Nguyen et al., 2025; Xu et al., 2024) and STIX bundle generation (Siracusano et al., 2023). Later studies moved from entity extraction to relation modeling, enabling the construction of cybersecurity knowledge graphs (Cheng et al., 2025; Huang & Xiao, 2024). However, these approaches remain narrow in scope: they typically operate on individual unstructured reports, target only limited task families, and lack systematic adaptation and evaluation across heterogeneous CTI sources like structured repositories.

**CTI Benchmarks.** Although LLM benchmarks have proliferated in general NLP, only a few have targeted the CTI domain. CTIBench (Alam et al., 2025) was a notable effort, evaluating LLMs on four tasks: root cause mapping, vulnerability severity prediction, attack technique extraction, and threat actor attribution. However, these tasks capture only a narrow slice of the CTI analysis landscape. CTIBench also evaluated only a few models under closed-book settings, relying solely on parametric knowledge without retrieval or augmentation. Its dataset was manually curated at small scale, making it difficult to extend to emerging threats. Another effort, SEvenLLM (Ji et al., 2024), introduced SEvenLLM-Bench, a bilingual multi-task dataset covering 28 CTI-related tasks (13 understanding and 15 generation). While the task count appears large, the scope remains narrow: the benchmark is restricted to unstructured CTI reports, focusing primarily on single-report extraction and summarization. Its evaluation is also limited to instruction-tuned small models ($\leq$14B parameters), excluding frontier LLMs that may exhibit stronger reasoning capabilities.

**LLM for Cybersecurity.** Recent studies have also explored the use of LLMs in a range of cybersecurity challenges outside CTI. PentestGPT (Deng et al., 2023) explores their role in penetration testing, showing that while LLMs can handle fundamental tasks and operate testing tools effectively,

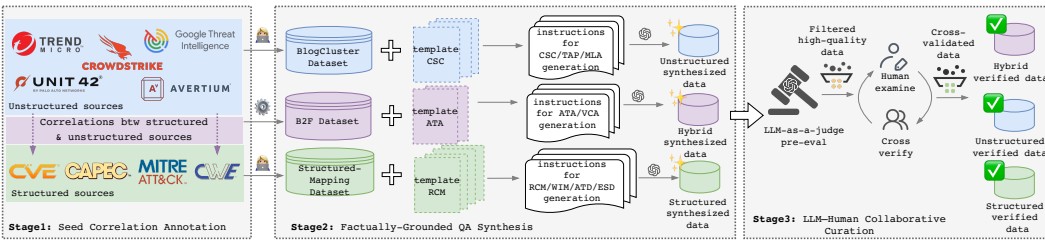

**Fig. 1:** Each CTI task in CTIARENA is created through a three-stage construction process. The task quality is controlled by the human-LLM collaboration.

they struggle with context retention and attention management. Beyond penetration testing, LLMs have also been investigated for vulnerability detection (Fang et al., 2024; Lu et al., 2024), software repair (Kulsum et al., 2024), fuzz testing (Xia et al., 2024; Meng et al., 2024), phishing and scam detection (Jiang, 2024; Lee et al., 2024), and DDoS detection (Li et al., 2024; Guastalla et al., 2023).

## 3  CTIARENA DESIGN

### 3.1  PRIMER ON CYBER THREAT INTELLIGENCE

Open-source CTI can be broadly categorized into structured and unstructured intelligence.

**Structured CTI** serves as a standardized knowledge base of vulnerabilities, weaknesses, and adversary behaviors, typically organized into three types: **vulnerability intelligence**, **weakness intelligence**, and **adversary behavior intelligence**. CVE (Program, 2025) catalogs vulnerabilities with identifiers, references, and severity scores. CWE (MITRE, 2025b) captures recurring weakness types to address root causes early in design. Adversary behavior is documented in CAPEC (MITRE, 2025a), which enumerates recurring attack patterns (e.g., SQL injection), and MITRE ATT&CK (MITRE, 2025c), which maps tactics and techniques observed in real intrusions across the attack lifecycle. CAPEC emphasizes application-level patterns, while ATT&CK captures operational adversary behaviors. Further details are provided in Appendix A.1.

**Unstructured CTI** refers to narrative intelligence in vendor reports and blogs (e.g., Trend Micro, Unit 42, Google Threat Intelligence). These sources describe adversary activities with rich context: actors, malware, campaigns, TTPs, IoCs, and mitigations, providing timely insights into emerging threats. Unlike structured CTI, their free-text format makes systematic management and cross-report correlation more challenging.

### 3.2  BENCHMARK CONSTRUCTION PIPELINE

Given the scale and diversity of CTI tasks, fully manual annotation is impractical, and naive LLM generation risks hallucination. We therefore adopt a three-stage LLM-based pipeline (illustrated in Fig. 1) that grounds data in authoritative references and enforces quality control throughout.

*Stage 1: Seed Correlation Annotation:* We collect high-quality correlations among CTI sources as ground truth for dataset generation. For structured CTI, we extract mappings from authoritative repositories (e.g., CVE→CWE, CWE→CAPEC, CAPEC→ATT&CK), which are designed to interoperate: CVEs record vulnerabilities, CWEs their root causes, CAPEC exploitation patterns, and ATT&CK adversary tactics and techniques. These are organized into the *StructuredMapping* dataset. For unstructured CTI, we cluster vendor reports by adversary type (threat actors, malware, campaigns) into the *BlogCluster* dataset, resolving aliases and verifying topic consistency (details in Appendix B). Finally, we construct the *B2F (Blog-to-Framework)* dataset by aligning key phrases in blogs with structured taxonomies. Two annotators independently highlight phrases and map them to authoritative entries, cross-check their results, and resolve disagreements with a senior adjudicator. Together, these seed datasets anchor our pipeline with authoritative references (see Section 3.3).

*Stage 2: Factually-Grounded QA Pair Synthesis:* We transform the annotated seed correlations into task-specific QA pairs using *template-constrained generation*. Each correlation is represented as a *subject–object* pair (e.g., CVE→CWE). Prompt templates then guide the LLM, specifying (i)

the task-specific instruction, (ii) the input entity (subject), (iii) the expected output entity (object), and (iv) the required QA format. By grounding prompts in verified correlations and constraining outputs with templates, the LLM produces QA pairs that are directly linked to authoritative sources, reducing hallucination and ensuring consistency across tasks. Representative prompt templates for all task families are provided in Appendix E.1.

*Stage 3: LLM–Human Collaborative Curation:* The final stage enforces multi-layered quality control on the synthesized QA pairs. We first implement an LLM-based judge with GPT-5 as the backbone, guided by a structured rubric (Appendix E.2) that evaluates factual correctness, grounding in source evidence, clarity of language, and consistency with the task definition. Samples flagged as low-confidence or inconsistent are discarded, while high-confidence outputs are retained for further validation. Next, two cybersecurity practitioners with over two years of hands-on CTI analysis experience independently answer the LLM-generated questions and cross-check their responses. Finally, a senior annotator with more than three years of professional CTI expertise conducts a final review to ensure that the retained QA pairs are factually accurate, unambiguous, and faithful to the intended task design. This collaborative design, which combines scalable LLM-based filtering with rigorous expert validation, consolidates the synthesized QA data into a final, verified benchmark dataset.

### 3.3 TASK DESIGN

#### 3.3.1 STRUCTURED CTI REASONING

Structured CTI reasoning tasks evaluate whether LLMs can infer correlations among security taxonomies (CVE, CWE, CAPEC, MITRE ATT&CK). For example, a CVE may stem from a CWE, be exploited via a CAPEC pattern, and map to an ATT&CK technique. Automating such reasoning reduces the manual burden on analysts and supports more timely, systematic defenses.

**CTI-RCM (Root Cause Mapping).** This task maps a vulnerability in CVE to its root cause in CWE. For example, CVE-2021-44228 (Log4Shell) maps to CWE-20 (Improper Input Validation). Identifying such root causes enables proactive defense by addressing weaknesses at design time.

**CTI-WIM (Weakness Instantiation Mapping).** This task links a weakness in CWE to a CVE that instantiates it. For example, CWE-79 (Improper Input Neutralization) is instantiated by CVE-2021-40444 in Microsoft Office, which allows malicious ActiveX controls via crafted documents. This enables defenders to assess the prevalence and impact of weaknesses across deployed systems, informing prioritization and patching strategies.

**CTI-ATD (Attack Technique Derivation).** This task maps an attack pattern in CAPEC to the corresponding tactics and techniques in MITRE ATT&CK. For instance, CAPEC-66 (SQL Injection) corresponds to ATT&CK technique T1190 (Exploit Public-Facing Application). This helps link abstract design-level attack classes to specific tactics and techniques that adversaries employ.

**CTI-ESD (Exploitation Surface Discovery).** This task links weaknesses in CWE to attack patterns in CAPEC, showing how a weakness can be exploited. For example, CWE-352 (Cross-Site Request Forgery) is associated with CAPEC-111 (CSRF Attack). This helps link software weakness to exploitation methods, enabling analysts to anticipate attack vectors and prioritize mitigations.

*Dataset.* We curated the **StructuredMapping Dataset**, which contains 500 high-quality mappings that explicitly connect entities across structured CTI frameworks (e.g., CVE→CWE, CWE→CVE, CAPEC→ATT&CK, CWE→CAPEC). Building on this resource, we applied our dataset construction pipeline (see Section 3.2) to derive the **SK (Structured Knowledge) Dataset**, comprising 371 QA pairs in total: 100 instances for CTI-RCM, 71 for CTI-WIM, 100 for CTI-ATD, and 100 for CTI-ESD. Data examples are provided in Appendix C.2.

#### 3.3.2 THREAT REPORT UNDERSTANDING

Unstructured CTI tasks evaluate whether LLMs can extract and synthesize adversary-centric insights from narrative threat reports. These tasks are crucial as reports provide timely, context-rich intelligence, yet their free-text format makes cross-source analysis challenging.

**CTI-CSC (Campaign Storyline Construction).** This task reconstructs the storyline of an attack campaign including exploited vulnerabilities, adversary tools and infrastructure, progression of ac-

tivities (e.g., lateral movement, persistence, exfiltration), and targeted industries or regions. For instance, separate reports might respectively describe a spear-phishing lure, an exploit of a public-facing service, and subsequent data exfiltration targeting financial institutions on the same compaign.

**CTI-TAP (Threat Actor Profiling).** This task construct the profile of a threat actor, including its known aliases, attributed malware/toolsets, observed TTPs, and targeted industries or regions. For example, reports about APT29 actor may separately mention credential harvesting, Cobalt Strike usage, or campaigns against government organizations.

**CTI-MLA (Malware Lineage Analysis).** This task trace the lineage of a malware family, including successive variants, code modifications, persistence or delivery mechanisms, reused components, and relationships to other families or toolsets. For example, multiple reports may describe successive variants of the Emotet malware, each introducing new persistence mechanisms or delivery vectors.

*Dataset & Corpus*. We collected 321 CTI reports from 35 major threat intelligence vendors, including Trend Micro (Trend Micro, 2025), Unit 42 (Palo Alto Networks, 2025), and Crowd-Strike (CrowdStrike, 2025); the full vendor list and report distribution are provided in Appendix A.2. Reports were categorized into threat actors, malware, and campaigns, and aggregated into clusters of reports sharing a *common adversary focus*. This yielded the **BlogCluster Dataset**, consisting of 50 expert-annotated clusters of CTI reports. Within each cluster, reports were ordered chronologically: the most recent report was used as the query input for evaluation tasks, while the remaining formed the retrieval corpus. Through our dataset construction pipeline Section 3.2, we derived the **UK (Unstructured Knowledge) Dataset**, including 60 instances for CTI-CSC, 60 for CTI-TAP, and 30 for CTI-MLA. Illustrative examples are provided in Appendix C.3.

### 3.3.3 STRUCTURED AND UNSTRUCTURED CTI MAPPING

Hybrid CTI tasks evaluate whether LLMs can bridge descriptions in threat reports with structured CTI taxonomies. This helps analysts connect free-text observations with standardized knowledge bases to analyze attack trends, identify underlying weaknesses, and determine possible mitigations.

**CTI-ATA (Attack Technique Attribution).** This task identifies attack behaviors in narrative reports that can be mapped to MITRE ATT&CK techniques. For example, if a report states that an adversary "used PowerShell scripts to download additional payloads", the model should map this behavior to ATT&CK technique T1059.001 (PowerShell).

**CTI-VCA (Vulnerability Catalog Attribution).** This task detects descriptions of vulnerable conditions in reports and correlates them with CWE entries that capture the underlying weakness. For instance, if a report mentions "unsanitized user input in a web form leading to code execution", the model should map this to CWE-20 (Improper Input Validation).

*Dataset*. We sampled 150 reports for paragraph-level analysis to identify latent weaknesses and ATT&CK techniques described in text. Two annotators conducted independent annotations, while a third annotator served as an arbiter to verify correctness, resolve disagreements, and filter out controversial or ambiguous cases. This yields *B2F (Blog-to-Framework) Dataset* Appendix C.1, containing 120 report-to-CWE mappings and 50 report-to-ATT&CK mappings. We applied our dataset construction pipeline (see Section 3.2) to derive the **HK (Hybrid Knowledge) Dataset**, comprising 120 instances for CTI-VCA and 50 for CTI-ATA. Data examples are provided in Appendix C.4.

## 4 EXPERIMENTS

### 4.1 EXPERIMENT SETUP

**Model Selection.** We evaluate ten state-of-the-art large language models, comprising both proprietary and open-source systems. Proprietary models include Claude-3.5-Haiku (Anthropic, 2024), Claude-Sonnet-4 (Anthropic, 2025), Gemini-2.5-Flash DeepMind (2025), Gemini-2.5-Pro (Google DeepMind, 2025), GPT-4o (OpenAI, 2024), and GPT-5 (OpenAI, 2025a). Open-source models include LLaMA-3-405B (Meta, 2024a), LLaMA-3-8B (Meta, 2024b), Phi-4 (Microsoft Research, 2024), and Qwen-3-235B (Qwen Team, 2025). To ensure strong instruction-following ability, we employ the official chat or instruction-tuned variants of all models.

**Baseline Implementation.** We evaluate LLMs under three major usage modes: closed-book reasoning, standard retrieval-augmented inference, and CTI-specific variants. Unless otherwise noted,

**Table II:** Comparison of LLM performance (F1-score, %) across CTIARENA task categories. "CB" denotes the closed-book setup, while "KW" denotes the best-performing knowledge-augmented setup (among all augmentation methods). Highest scores for open-source and proprietary models are marked with underline and **bold**, respectively.

| Model | Structured Tasks | | | | | | | | Unstructured Tasks | | | Hybrid Tasks | | | |
| --- | --- | --- | --- | --- | --- | --- | --- | --- | --- | --- | --- | --- | --- | --- | --- |
| | RCM | | WIM | | ATD | | ESD | | CSC | TAP | MLA | ATA | | VCA | |
| | CB | KW | CB | KW | CB | KW | CB | KW | KW | KW | KW | CB | KW | CB | KW |
| *Open-source LLMs* | | | | | | | | | | | | | | | |
| LLaMA-3-405B | 0.12 | 0.98 | 0.03 | 0.98 | 0.03 | 0.98 | 0 | 1 | 0.67 | 0.65 | 0.32 | 0.42 | 0.64 | 0.58 | 0.63 |
| LLaMA-3-8B | 0.01 | 0.89 | 0 | 1 | 0.01 | 0.96 | 0 | 1 | 0.59 | 0.46 | 0.38 | 0.10 | 0.40 | 0.13 | 0.28 |
| Phi-4 | 0.04 | 0.95 | 0 | 1 | 0.01 | 0.91 | 0 | 1 | 0.62 | 0.76 | 0.40 | 0.24 | 0.36 | 0.30 | 0.49 |
| Qwen-3-235B | 0.04 | 1 | 0 | 1 | 0.04 | 1 | 0 | 1 | 0.71 | 0.71 | 0.41 | 0.40 | 0.54 | 0.58 | 0.65 |
| *Proprietary LLMs* | | | | | | | | | | | | | | | |
| Claude-3.5-Haiku | 0.05 | 0.93 | 0.02 | **1** | 0 | 0.97 | 0 | **1** | 0.55 | 0.58 | 0.44 | 0.50 | 0.46 | 0.48 | 0.49 |
| Claude-Sonnet-4 | **0.24** | **0.99** | **0.15** | **1** | 0.01 | **0.99** | 0.01 | **1** | 0.55 | 0.56 | **0.48** | 0.46 | 0.64 | 0.71 | 0.73 |
| Gemini-2.5-Flash | 0 | 0.94 | 0.01 | 0.96 | 0.06 | **0.99** | 0 | **1** | 0.69 | 0.61 | 0.44 | 0.56 | 0.54 | 0.53 | 0.55 |
| Gemini-2.5-Pro | 0.03 | 0.92 | 0.03 | 0.98 | **0.09** | 0.96 | 0 | **1** | 0.61 | **0.79** | 0.36 | 0.58 | 0.70 | 0.63 | 0.57 |
| GPT-4o | 0.03 | 0.97 | 0 | 0.96 | 0.01 | **0.99** | 0 | **1** | 0.66 | 0.67 | 0.40 | 0.48 | 0.63 | 0.60 | 0.69 |
| GPT-5 | 0.09 | 0.98 | 0.05 | 0.99 | 0.06 | 0.95 | **0.01** | **1** | **0.72** | 0.66 | 0.39 | **0.64** | **0.76** | **0.83** | **0.90** |

all vanilla RAG baselines use text-embedding-3-large (OpenAI, 2025b) for semantic similarity computation, with the top-$k = 5$ results above a similarity threshold of 0.5 retained as context. These settings were chosen after exploratory trials as they yielded the most stable performance across tasks.

For ***structured tasks***, we consider (i) *closed-book*, where models rely solely on parametric knowledge, and (ii) *inference-time knowledge injection*, where identifiers from structured enumerations (e.g., CVE IDs) are detected via regular expressions and the corresponding official entries are directly injected into the prompt. We exclude vanilla RAG for this category, as structured CTI entries contain minimal descriptive text, making embedding-based retrieval ineffective and unnecessary.

For ***hybrid tasks***, we evaluate (i) *closed-book*; (ii) *vanilla RAG*, where queries are matched against the "description" fields of CVEs, CWEs, ATT&CK techniques, and CAPEC patterns; and (iii) *query-expanded RAG*, a CTI-specific variant. The motivation stems from the observation that generic embeddings often fail to align long narrative queries with domain-specific taxonomies. To address this, we decompose inputs into atomic CTI behaviors (e.g., tactics, techniques, or weakness symptoms) and perform retrieval separately for each, before aggregating the top-5 candidates for inference. This "divide-and-conquer" strategy closes the gap between narrative phrasing and structured taxonomy terms.

For ***unstructured tasks***, we first evaluate (i) *vanilla RAG*, where the input report is embedded and used to retrieve semantically similar reports from the corpus. However, vanilla RAG struggles when different vocabularies are used to describe the same adversary or malware. To address this, we introduce (ii) *CSKG-guided RAG*, which leverages a curated Cyber Security Knowledge Graph (CSKG) constructed with CTINexus (Cheng et al., 2025) under default settings. The CSKG encodes security entities (e.g., actors, malware, vulnerabilities, IoCs) and their relations, enabling reports to be linked through shared entities. For retrieval, entities are extracted from the query report, candidate reports with overlapping entities are identified, and those with an entity-overlap rate $\geq 0.6$ are retained; the top-$k = 5$ reports are then provided as evidence.

**Evaluation Protocol.** Given the heterogeneity of task formats, we adopt different evaluation approaches for each category. For **structured tasks** (RCM, WIM, ATD, ESD) and **hybrid tasks** (VCA, ATA), we use exact string matching with regex-based normalization to compare model outputs against ground-truth answers and compute accuracy. For **unstructured tasks** (MLA, TAP, CSC), the expected answers are open-ended and may vary in phrasing or level of detail. To handle this diversity, we employ GPT-5 as an automatic judge guided by a structured rubric (see Appendix E.2) to score predictions against reference answers at the bullet level, followed by human verification to ensure scoring reliability. In addition, for structured and hybrid tasks, we record and analyze model reasoning traces to support detailed error diagnosis and interpretability of results.

## 4.2 RESULT ANALYSIS

Table II summarizes the performance of all evaluated LLMs under closed-book (CB) and knowledge-augmented (KW) settings. Beyond results, three noteworthy patterns emerge:

**Table III:** Performance of LLMs on **unstructured tasks** under different baseline settings: Vanilla (standard RAG) and CSKG (CSKG-guided RAG over the unstructured corpus). The **best** values are highlighted.

| Model | CSC | | TAP | | MLA | |
|---|---|---|---|---|---|---|
| | Vanilla | CSKG | Vanilla | CSKG | Vanilla | CSKG |
| LLaMA–3–405B | 0.562 | 0.667 | 0.430 | 0.652 | 0.312 | 0.323 |
| LLaMA–3–8B | 0.476 | 0.592 | 0.462 | 0.424 | 0.258 | 0.379 |
| Phi–4 | 0.509 | 0.617 | 0.553 | 0.757 | 0.404 | 0.355 |
| Qwen–3–235B | 0.663 | **0.712** | 0.575 | 0.709 | 0.303 | 0.407 |
| GPT–5 | **0.721** | 0.671 | **0.583** | 0.663 | 0.362 | 0.394 |
| GPT–4o | 0.656 | 0.660 | 0.579 | 0.671 | 0.355 | 0.393 |
| Gemini–2.5–Flash | 0.570 | 0.694 | 0.540 | 0.608 | 0.320 | 0.441 |
| Gemini–2.5–Pro | 0.612 | 0.609 | 0.528 | **0.789** | 0.328 | 0.361 |
| Claude–3.5–Haiku | 0.435 | 0.548 | 0.476 | 0.584 | **0.443** | 0.410 |
| Claude–Sonnet–4 | 0.476 | 0.547 | 0.506 | 0.562 | 0.408 | **0.478** |

**Table IV:** Performance of LLMs on **hybrid tasks** under different baseline settings: Closed-book, Vanilla (standard RAG), and Expansion (RAG with query expansion). The **best** values are highlighted.

| Model | ATA | | | VCA | | |
|---|---|---|---|---|---|---|
| | Closebook | Vanilla | Expansion | Closebook | Vanilla | Expansion |
| LLaMA–3–405B | 0.575 | 0.558 | 0.625 | 0.420 | 0.540 | 0.640 |
| LLaMA–3–8B | 0.133 | 0.275 | 0.283 | 0.100 | 0.220 | 0.400 |
| Phi–4 | 0.308 | 0.375 | 0.492 | 0.240 | 0.480 | 0.360 |
| Qwen–3–235B | 0.583 | 0.575 | 0.650 | 0.400 | 0.560 | 0.540 |
| GPT–5 | **0.825** | **0.742** | **0.900** | **0.640** | 0.600 | **0.760** |
| GPT–4o | 0.600 | 0.642 | 0.692 | 0.480 | 0.580 | 0.620 |
| Gemini–2.5–Flash | 0.525 | 0.467 | 0.550 | 0.560 | 0.600 | 0.540 |
| Gemini–2.5–Pro | 0.625 | 0.558 | 0.567 | 0.580 | **0.740** | 0.700 |
| Claude–3.5–Haiku | 0.467 | 0.483 | 0.492 | 0.500 | 0.520 | 0.460 |
| Claude–Sonnet–4 | 0.708 | 0.667 | 0.725 | 0.460 | 0.580 | 0.640 |

**Table V:** Average retrieval performance on dynamic and hybrid tasks.

| Dynamic | | | | | | Hybrid (ATA) | | | | | | Hybrid (VCA) | | | | | |
|---|---|---|---|---|---|---|---|---|---|---|---|---|---|---|---|---|---|
| Vanilla | | | CSKG | | | Vanilla | | | Expansion | | | Vanilla | | | Expansion | | |
| Prec | Rec | F1 | Prec | Rec | F1 | Prec | Rec | F1 | Prec | Rec | F1 | Prec | Rec | F1 | Prec | Rec | F1 |
| 0.405 | 0.793 | 0.534 | 0.500 | 0.854 | 0.615 | 0.102 | 0.279 | 0.135 | 0.310 | 0.433 | 0.349 | 0.177 | 0.640 | 0.249 | 0.330 | 0.620 | 0.413 |

**(1) Structured task performance saturates once knowledge is provided.** Results in Table II show that structured CTI tasks remain highly challenging in the closed-book setting, with the best scores reaching only 0.24, 0.15, 0.99, and 0.01 on RCM, WIM, ATD, and ESD, respectively. These correlations are explicitly defined in authoritative, community-maintained repositories such as NVD for CVE and MITRE for CWE, CAPEC, and ATT&CK, yet most LLMs fail to internalize them as parametric knowledge, leading to frequent hallucinations. This weakness arises because mappings like CVE↔CWE or CAPEC↔ATT&CK represent long-tail enumerated knowledge that is rarely absorbed during pretraining. Once authoritative entries are injected or retrieved, reasoning becomes straightforward and accuracy quickly saturates. This shows that the difficulty of structured CTI tasks stems less from complex inference and more from whether models are supplied with the correct references. We also observe that open-source models lag behind proprietary ones in the closed-book mode, suggesting broader coverage of security-related content in proprietary pretraining. Crucially, the gap disappears once external knowledge is provided: with authoritative evidence available, all models achieve near-perfect accuracy. Hence, structured CTI reasoning hinges on reliable grounding in external repositories, and simple knowledge augmentation is sufficient to close the gap.

**(2) Hybrid tasks pose challenges in knowledge retrieval and grounding.** Table IV highlights several consistent patterns. First, knowledge augmentation improves performance over closed-book baselines, but gains vary by model strength: stronger LLMs exploit retrieved content effectively, while weaker ones struggle, widening the gap. Second, *query-expanded RAG*, which reformulates inputs into fine-grained CTI facets (e.g., tactics, techniques, or affected components), consistently outperforms vanilla RAG by better aligning queries with the structure of target taxonomies. This yields substantial improvements, particularly for GPT-5 (ATA: $0.742{\rightarrow}0.900$; VCA: $0.600 \rightarrow 0.760$). Third, proprietary LLMs achieve the strongest results, with GPT-5 far ahead of open-source counterparts, while smaller models such as LLaMA-3-8B and Phi-4 remain weak even under retrieval augmentation. Finally, ATA tasks appear consistently easier than VCA: ATT&CK techniques are expressed in concrete behavioral terms (e.g., command execution, credential dumping), which map directly to textual cues, whereas CWE weaknesses encode abstract design flaws (e.g., improper validation) that must be inferred from indirect symptoms. Hence, VCA demands deeper security knowledge and more semantic abstraction, making it the harder grounding problem.

**(3) Unstructured tasks are constrained by cross-report synthesis rather than retrieval alone.** Unstructured tasks (CSC, TAP, MLA) yield the lowest accuracy, even under knowledge-augmented setups. This difficulty arises because evidence is scattered across heterogeneous reports that differ in style, granularity, and alias usage: the same actor may appear under multiple names, and malware variants may be described with inconsistent technical detail. As a result, embedding-based vanilla

RAG often fails, semantic similarity alone misses correlations when surface forms diverge. Within these tasks, CSC is relatively easier since campaign timelines are often described explicitly, while MLA is hardest because reconstructing malware evolution requires integrating incremental changes across reports. Entity-centric retrieval with *CSKG-guided RAG*, which links reports through overlapping entities, produces smaller but more precise evidence sets and consistently outperforms vanilla RAG. These results highlight that in unstructured CTI, precision on the right entities matters more than retrieval breadth, and that the central bottleneck lies in synthesizing fragmented evidence across reports even when relevant documents are retrieved.

### 4.3 ERROR ANALYSIS

We analyze failure patterns that reveal why CTI reasoning remains difficult for current LLMs.

**Semantic drift from noisy retrieval.** In VCA tasks, naive integration of retrieved passages sometimes amplifies noise. For example, when retrieval returned documents that *appeared similar in wording* (e.g., discussions of password policies) rather than those describing the actual weakness (e.g., CWE-308: Use of Single-Factor Authentication), models such as Claude-3.5-Haiku and Gemini-2-Pro misclassified CWE-308 as CWE-521 or CWE-307. We define this as *semantic drift*: retrieved evidence that is textually close to the query but corresponds to a different security concept. Around 8% of VCA cases showed such drift, reducing accuracy from 63% to 55%. This highlights a core limitation of semantic-similarity retrieval: passages that look linguistically relevant can still misalign conceptually, steering models toward the wrong taxonomy entry. Robust CTI retrieval therefore requires distinguishing superficial textual resemblance from evidence that is conceptually faithful to the intended security category.

**Resistance to leveraging correct evidence.** Smaller models such as LLaMA-3-8B sometimes retrieved the correct evidence but failed to use it, instead reverting to incorrect default associations. For instance, in CWE-521 cases, the model predicted unrelated weaknesses even though the correct CWE description was available in the retrieved context. Similarly, Gemini-2.5-Flash repeatedly mapped CWE-807 to CWE-451 despite having the correct supporting evidence retrieved. This "retrieval-but-not-used" phenomenon accounted for 15% of VCA errors. While this behavior reduces susceptibility to noisy retrieval, it also prevents models from correcting outdated or misaligned internal knowledge.

**Instability of tailored approaches in smaller models.** In ATA tasks, smaller open models such as Phi-4 showed fluctuating performance depending on the retrieval strategy. Error analysis reveals that these models often struggled to follow decomposition instructions, producing incorrect splits of the narrative and thereby retrieving noisy documents. As a result, Phi-4 lost 12% accuracy with query-expanded RAG compared to vanilla RAG. This suggests that retrieval strategies that improve coverage for stronger models can instead destabilize weaker ones, showing a tradeoff between retrieval complexity and model robustness.

**Implicit boundary between internal knowledge and retrieval grounding.** This subtle failure mode was identified only through close inspection of reasoning traces. In some cases, models produced the correct answer using internal knowledge, but simultaneously claimed to ground their prediction in retrieved references, even when the ground-truth evidence was not among the retrieved candidates. This creates an illusion of faithful reasoning while in fact the justification is fabricated. We observed such "unsupported correctness" in 3% of predictions overall, with GPT-4o showing the highest rate at 7%. This pattern reflects an implicit boundary between parametric knowledge and retrieval-based grounding. Although the final answer may be correct, unverifiable justifications undermine trust in CTI applications.

## 5 CONCLUSION

We introduced CTIARENA, the first benchmark for evaluating LLMs on heterogeneous, multi-source CTI. Our dataset contains 691 high-quality QA pairs grounded in authoritative CTI sources, enabling rigorous and realistic evaluation. Experiments reveal both the limitations of current LLMs and the promise of retrieval-augmentation strategies, establishing CTIARENA as a catalyst for future research on domain-tailored methods and the development of next-generation CTI copilots.

# 6 ETHICS STATEMENT

This work complies with the ICLR Code of Ethics. We collect our data only from open-source CTI platforms that contain no personal or sensitive data, and we are committed to transparency in our methodology for annotating the dataset. No human subjects, privacy concerns, or dual-use risks are involved. We see no foreseeable societal harms from this research and affirm our commitment to ethical and responsible conduct.

# 7 REPRODUCIBILITY STATEMENT

Upon publication, we will release the complete CTIARENA dataset, including all questions, answers, references, and the supporting corpus. We will also provide a comprehensive codebase covering all experimental implementations and evaluation metrics to support reproducibility. A preview version of the codebase is currently available at `https://anonymous.4open.science/r/CTIArena-04FF`. By making these resources publicly accessible, we aim to facilitate transparent evaluation and foster future research on LLMs for CTI.

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

# A APPENDIX

## A.1 SECURITY KNOWLEDGE FRAMEWORKS

- **CVE (Common Vulnerabilities and Exposures).** CVE provides a standardized catalog of publicly disclosed vulnerabilities. Each entry is assigned a unique CVE identifier and a short description, enabling consistent tracking across security advisories, vendors, and tools. CVE serves as the industry baseline for vulnerability identification and reporting.
- **CWE (Common Weakness Enumeration).** CWE generalizes beyond individual vulnerabilities by categorizing recurring software and hardware weakness patterns. Each CWE entry defines the weakness, describes its consequences, and provides potential mitigations. CWE thus supports reasoning about the root causes underlying CVE instances.
- **CAPEC (Common Attack Pattern Enumeration and Classification).** CAPEC documents reusable adversarial attack patterns that exploit weaknesses (CWE). Each CAPEC entry specifies the attack mechanics, required conditions, and potential impacts. By bridging weaknesses with exploitation strategies, CAPEC facilitates systematic threat modeling.
- **MITRE ATT&CK.** ATT&CK is an empirically grounded knowledge base of adversarial tactics, techniques, and procedures (TTPs) observed in real-world intrusions. Organized into a matrix, ATT&CK captures how attackers operate across the intrusion lifecycle, from initial access to impact. It has become the de facto framework for threat hunting, detection, and evaluation.

## A.2 DISTRIBUTION OF REPORTS

Table VI summarizes the top 10 distribution of blog sources contributing to our dataset. The ranking reveals that a small number of highly active security news outlets (e.g., The Hacker News, BleepingComputer, and SecurityWeek) dominate coverage, while contributions from industry blogs (e.g., Unit42 and Microsoft) ensure representation of vendor-driven intelligence. The vertical ellipsis indicates additional long-tail sources not explicitly listed. In total, our corpus aggregates 321 unique blog entries from 35 popular threat intelligence vendors, with 63% of them originating after 2020.

**Table VI:** Top-10 distribution of blog sources.

| Rank | Source | Count |
|------|--------|-------|
| 1 | thehackernews | 31 |
| 2 | BleepingComputer | 29 |
| 3 | securityweek | 28 |
| 4 | darkreading | 27 |
| 5 | Unit42 | 26 |
| 6 | AVERTIUM | 20 |
| 7 | welivesecurity | 20 |
| 8 | trendmicro | 19 |
| 9 | threatPost | 19 |
| 10 | Microsoft | 18 |
| ⋮ | ⋮ | ⋮ |
| **Total** | | **321** |

# B BLOGCLUSTER ANNOTATION PROCEDURE

The *BlogCluster* dataset organizes unstructured CTI reports into clusters of adversaries, covering three categories: threat actors, malware families, and campaigns. The construction procedure is as follows.

**Step 1: Topic identification and candidate clustering.** Annotators first read each report to identify its primary topic (threat actor, malware, or campaign). The extracted keyword is then used to search

across the corpus, retrieving other reports where the same entity appears as the main subject. These are grouped into candidate clusters by adversary type.

**Step 2: Alias resolution and metadata reconciliation.** Entity names often differ across vendors (e.g., "APT28," "Fancy Bear," "Sofacy"), so annotators normalize aliases to ensure that all references to the same adversary are grouped together. In addition, inconsistencies in metadata (e.g., labels, families, or campaign names) are reconciled by reviewing entity descriptions, activity timelines, and referenced indicators.

**Step 3: Validation and cross-verification.** Each candidate cluster is manually inspected to verify topical consistency, with off-topic or ambiguous reports removed. To ensure reliability, two annotators independently perform validation and reconciliation, after which a senior adjudicator reviews disagreements and confirms the final clusters. Only clusters with at least two reports from different vendors or perspectives are retained in the *BlogCluster* dataset.

This procedure ensures that *BlogCluster* captures accurate, multi-source coverage of adversaries while resolving vendor-specific variations and maintaining strict quality control.

## C  ADDITIONAL DATA EXAMPLES

### C.1  B2F DATASET

**Table VII:** Mapping of security blogs to CWE entries in the B2F dataset.

| ID | Blog Title | CWE Mappings | Vulnerability Summaries |
|---|---|---|---|
| 1 | Microsoft Zero-Day Vulnerabilities | CWE-693: Protection Mechanism Failure; CWE-400: Resource Exhaustion | CVE-2023-36884 bypassed Mark-of-the-Web protections, enabling RCE; CVE-2023-38180 caused .NET DoS via resource exhaustion. |
| 2 | BlackCat Ransomware & Triple Extort | CWE-693: Protection Mechanism Failure; CWE-284: Improper Access Control | BlackCat disables security/backup processes, bypassing defenses; weak privilege isolation allows termination of defenses and lateral movement. |
| 3 | Silent Skimming | CWE-502: Deserialization of Untrusted Data; CWE-1104: Use of Unmaintained Components; CWE-434: Dangerous File Upload; CWE-94: Code Injection; CWE-269: Privilege Management; CWE-284: Access Control; CWE-200: Info Exposure; CWE-778: Insufficient Logging | Exploited Telerik deserialization (CVE-2019-18935) for RCE; outdated component allowed initial access; malicious uploads (DLLs/webshells); arbitrary code injection; Potato tools escalated privileges; weak access controls enabled lateral moves; payment data exfiltrated; inadequate logging allowed long-term persistence. |
| 4 | Akira Ransomware | CWE-668: Exposure to Wrong Sphere; CWE-307: Excessive Authentication Attempts; CWE-287: Improper Authentication; CWE-521: Weak Passwords; CWE-522: Unprotected Credentials; CWE-284: Access Control; CWE-693: Protection Mechanism Failure; CWE-1104: Unmaintained Components; CWE-200: Info Exposure; CWE-319: Cleartext Transmission | Publicly exposed RDP/ESXi enabled brute force; missing lockout on RDP; absence of MFA allowed credential use; weak passwords cracked; credentials dumped from memory; AD/DC misconfigs enabled persistence; defenses disabled; outdated ESXi exploited; exfiltrated data leaked; FTP exfiltration risked plaintext exposure. |
| 5 | CACTUS and 3AM Ransomware | CWE-22: Path Traversal; CWE-918: SSRF; CWE-862: Missing Authorization; CWE-306: Missing Authentication; CWE-269: Privilege Management; CWE-693: Protection Mechanism Failure; CWE-668: Exposure to Wrong Sphere; CWE-829: Untrusted Functionality Inclusion | Qlik CVE-2023-41266 allowed unauthorized HTTP requests; CVE-2023-41265 SSRF enabled escalation; unauthorized actions executed without checks; sensitive operations without authentication; privilege escalation via tunneling; incomplete patch bypassed; internet-exposed Qlik broadened attack surface; malvertising embedded untrusted functionality. |
| 6 | Atlassian Confluence RCE Exploitation | CWE-1336: Improper Neutralization of Special Elements in Template Engine; CWE-94: Code Injection; CWE-20: Improper Input Validation; CWE-1104: Unmaintained Components | SSTI in Confluence enabled unauthenticated RCE; injected templates led to arbitrary code execution; improper validation allowed injection; outdated Confluence left systems exposed. |
| 7 | Chrome Zero-Day 2024 Exploitation | CWE-125: Out-of-bounds Read; CWE-787: Out-of-bounds Write; CWE-843: Type Confusion | CVE-2024-0519 allowed buffer over-read exposing memory; CVE-2024-0517 enabled out-of-bounds writes leading to corruption/RCE; CVE-2024-0518 type confusion caused memory corruption and possible code execution. |

## C.2 STRUCTURED DATASET

**Table VIII:** Example questions and answers in the Structured dataset.

| Task | Question | Answer |
|---|---|---|
| RCM | Which software weakness corresponds to CVE-2023-38327, and why is that the root cause of this vulnerability? | CWE-203. CVE-2023-38327 causes user enumeration via response differences, matching CWE-203: observable discrepancy exposing state information. |
| RCM | Which CWE maps to CVE-2024-10644 and why is that weakness the root cause of this remote code execution issue? | CWE-94. CVE-2024-10644 causes code injection via unneutralized external input, matching CWE-94: constructs code segments using externally-influenced input without proper neutralization. |
| RCM | Which CWE is the root cause of CVE-2024-11824, and why does this vulnerability map to that weakness? | CWE-79. CVE-2024-11824 causes stored XSS by not neutralizing input, matching CWE-79: improper neutralization of input during web page generation. |
| WIM | Which CVE instantiates CWE-192 (Integer Coercion Error), and why does that CVE demonstrate an integer coercion error? | CVE-2022-2639. Demonstrates integer coercion error via incorrect size reservation, instantiating CWE-192: type casting or truncation flaws. |
| WIM | Which CVE instantiates CWE-1426 (Improper Validation of Generative AI Output), and why does it demonstrate this weakness? | CVE-2024-3402. Demonstrates improper validation of generative AI output, instantiating CWE-1426: insufficient validation of AI-generated outputs. |
| WIM | Which CVE instantiates CWE-1231 (Improper Prevention of Lock Bit Modification), and why does that vulnerability demonstrate this weakness? | CVE-2017-6283. Demonstrates improper prevention of lock bit modification, instantiating CWE-1231: uses a trusted lock bit but allows modification after setting. |
| ATD | Which MITRE ATT&CK technique maps to CAPEC-141 (Cache Poisoning)? Explain why that technique corresponds to this attack pattern. | T1557.002. Corresponds to ARP cache poisoning, mapping CAPEC-141: attacker places incorrect or harmful material in cache. |
| ATD | Which MITRE ATT&CK technique maps to CAPEC-2 Inducing Account Lockout? Explain why this technique corresponds to this attack pattern. | T1531. Corresponds to user account access removal, mapping CAPEC-2: attacker leverages throttling mechanism to lock out legitimate users. |
| ATD | Which MITRE ATT&CK technique maps to CAPEC-25 (Forced Deadlock)? Explain why that technique corresponds to this attack pattern. | T1499.004. Exploits deadlock causing denial of service, mapping CAPEC-25: adversary triggers and exploits a deadlock condition. |
| ESD | Which CAPEC attack pattern exploits CWE-125's out-of-bounds read vulnerability? Explain why this pattern can exploit this weakness. | CAPEC-540. Exploits out-of-bounds read, targeting CWE-125: product reads data past end or before beginning of buffer. |
| ESD | Which CAPEC attack pattern exploits CWE-190 (Integer Overflow or Wraparound)? Explain why that pattern can exploit this weakness. | CAPEC-92. Exploits integer overflow, targeting CWE-190: calculations exceed storage bounds. |
| ESD | Which CAPEC attack pattern exploits CWE-147? Explain why that CAPEC pattern can exploit the improper neutralization of input terminators vulnerability. | CAPEC-460. Exploits parameter pollution, targeting CWE-147: product receives input from upstream, does not neutralize special elements interpreted as input terminators. |

## C.3 Unstructured dataset

**Table IX:** Example questions and answers in the Unstructured dataset.

| Task | Question | Answer |
|------|----------|--------|
| CSC | What malware families, threat groups, or vulnerability IDs appear consistently across these reports? **{Report Contents}** | MalwareTools: PowerShell RAT (server.ps1), Cobalt Strike beacons; Threat Actor Groups: Silent Skimmer; Vulnerability IDs: CVE-2019-18935 |
| CSC | What are the primary target industries or regions mentioned across these reports? **{Report Contents}** | Regions: Taiwan; China; Hong Kong; Japan; South Korea; broader Eastern Asia; overseas Taiwanese diaspora; Sectors/Industries: government and research organizations (defense and policy); journalists and dissidents; business and technology leaders. |
| CSC | What is the time span of this DPRK "Contagious Interview" campaign based on the earliest and latest report dates? **{Report Contents}** | Earliest report date: December 2023 (first documentation of the Ferret macOS malware family). Latest report date: April 23, 2025 (FBI seizure of blocknovas.com). |
| TAP | What is the canonical threat actor name, resolving any aliases mentioned across these reports? **{Report Contents}** | Canonical Threat Actor Name: APT42. Aliases Resolved: APT35, TA453, Mint Sandstorm. Affiliation: IRGC-linked Iranian state-sponsored actor. Primary TTPs: spear-phishing with malicious RTF/PDF attachments, zero-day SQL injection, ChainDoor RAT, custom Android spyware, Mimikatz credential harvesting, MFA interception, steganographic data exfiltration, coin-miner payloads. Main Targets: government ministries, critical infrastructure, officials, policymakers, journalists, academics, dissidents. Key Insight: multi-stage espionage-to-ransom capability with emphasis on credential theft, lateral movement, and large-scale data exfiltration. |
| TAP | What are the primary target regions and sectors mentioned across these reports for BladedFeline's cyberespionage activities? **{Report Contents}** | Primary target regions: Kurdish Regional Government in northern Iraq, Iraqi government networks, and a telecommunications provider in Uzbekistan. Primary target sectors: oil and energy infrastructure, government and diplomatic communications, and telecom services. Key intelligence objectives: exfiltration of diplomatic and financial data tied to regional oil reserves and Western relations. |
| MLA | What are all the distinct malware variant names mentioned across these reports? **{Report Contents}** | LockBit, CL0P, CACTUS, 3AM, FakeUpdates, Dridex, WastedLocker, PayloadBin, Hades. |
| MLA | Across these reports, what is the single most notable new capability that appeared in later variants compared to earlier ones? **{Report Contents}** | Early variants: phishing-delivered HTML attachments to drop Get2 loader, deploy RATs (SDBOT, FlawedAmmyy), Cobalt Strike, TinyMet reverse shells, and DEWMODE web shell for exfiltration. Mid-stage: exploitation of MOVEit Transfer SQL injection zero-day with Lemurloot web shell to manipulate databases, exfiltrate records and Azure settings. Later variants: shifted data-leak operations from Tor-hosted downloads to decentralized BitTorrent distribution (magnet links, DHT trackers) seeding terabytes of stolen data. Most Notable New Capability: Decentralized BitTorrent-based data leak distribution. |
| MLA | What is the most notable new capability that appeared in later variants compared to earlier ones? **{Report Contents}** | Introduction of a zero-day Windows elevation-of-privilege exploit (CVE-2025-29824), enabling SYSTEM-level access and enhanced persistence and lateral movement. |

## C.4 Hybrid Dataset

**Table X:** Example questions and answers in the Hybrid dataset.

| Task | Question | Answer |
|---|---|---|
| VCA | I recently read in a security blog: "Lack of lockout/rate-limiting on database or remote access logins enabled repeated guessing until credentials were obtained." Please provide: 1) the CWE category, 2) an explanation of the mapping, and 3) potential mitigations. | 1) CWE-307: Improper Restriction of Excessive Authentication Attempts. 2) Mapping Explanation: No lockout or rate-limiting enables unlimited guessing, directly addressed by CWE-307. 3) Mitigations: disconnect after a few failed attempts; implement timeouts; lock accounts after repeated failures; require CAPTCHA or equivalent. |
| VCA | I recently read in a security blog: "Database services were directly exposed to the internet, allowing external adversaries to hijack them." Please provide: 1) the CWE category, 2) mapping explanation, and 3) modes of introduction. | 1) CWE-668: Exposure of Resource to Wrong Sphere. 2) Mapping Explanation: Exposing database services to the public internet aligns with CWE-668. 3) Modes of Introduction: Architecture and Design; Implementation (security tactic misapplied); Operation. |
| VCA | I recently read in a security blog: "Workflow caused Office/MSHTML or Windows to load and execute ActiveX/DLL components from attacker-controlled servers or ISOs." Please provide: 1) the CWE category, 2) explanation, and 3) detection methods. | 1) CWE-829: Inclusion of Functionality from Untrusted Control Sphere (noted under id 910 with cwe_id 829). 2) Mapping Explanation: Loading untrusted ActiveX/DLL components directly aligns with CWE-829. 3) Detection Methods: Automated/Manual Static Analysis (binary and source), Dynamic Analysis with sandboxing, and Architecture or Design Review. |
| ATA | I recently read in a security blog: "Cuba ransomware operators were infiltrating networks by encrypting files using the ".cuba" extension." Please provide: 1) the MITRE ATT&CK category that maps to this behavior, 2) an explanation of the mapping between the ransomware behavior and the technique, and 3) which operating systems and platforms are affected? | 1) MITRE ATT&CK Category: Data Encrypted for Impact (T1486). 2) Mapping Explanation: Encrypting files to disrupt business operations aligns with T1486. 3) Affected Platforms: Linux, macOS, Windows, IaaS, ESXi. |
| ATA | I recently read in a security blog: "Once users open the malicious document, a new version of a .Net credential stealer is loaded via Follina, stealing credentials from Edge and Chrome browsers." Please identify the MITRE ATT&CK category and affected operating systems. | 1) MITRE ATT&CK Category: T1555.003 - Credentials from Web Browsers. 2) Mapping Explanation: Malicious document triggers credential stealer aligned with T1555.003. 3) Applicable Platforms: Linux, macOS, Windows. |
| ATA | I recently read in a security blog: "In May 2023, ZScaler detailed CryptNet ransomware. Operators claim to exfiltrate data prior to encryption. Like CryptNet, Mallox also uses double extortion." Which MITRE ATT&CK technique category maps to this behavior, how does it map, and which platforms are affected? | 1) MITRE ATT&CK Technique: T1486 - Data Encrypted for Impact. 2) Mapping Explanation: CryptNet and Mallox exfiltrate data before encrypting, consistent with T1486. 3) Applicable Platforms: Linux, macOS, Windows, IaaS, ESXi. |
| ATA | I recently read in a security blog: "For Iraqi government victims, ESET suspects the group exploited a flaw in an internet-facing web server enabling web shell deployment." Which MITRE ATT&CK technique does this correspond to, how does this behavior map to it, and where in the attack lifecycle should we expect this behavior? | 1) T1190 - Exploit Public-Facing Application 2) Mapping Explanation: The adversary leveraged a vulnerability in an internet-facing web server to deploy a web shell, which directly aligns with exploiting a public-facing application as defined in T1190. 3) Kill Chain Phase: Initial Access |

## D  THE USE OF LARGE LANGUAGE MODELS (LLMS)

This work investigates the capabilities of LLMs in CTI analysis tasks. LLMs were also used in constructing and verifying the benchmark dataset. Beyond this, their use was limited to writing assistance for language polishing and stylistic refinement. All technical content, formulations, experimental designs, and conceptual contributions were developed by the authors. Importantly, LLMs were not involved in research ideation.

## E  PROMPT TEMPLATE

### E.1  PROMPT IN DATASET GENERATION

Attack Technique Attribution (ATA) belongs to Dynamic CTI tasks. In this prompt, variables enclosed in curly braces (e.g., {SOURCE_NODE}, {TARGET_NODE}, {ASPECT}) are placeholders that are dynamically replaced at runtime.

---

**Generation Prompt for ATA**

INSTRUCTION
You are an Attack Technique Attribution (ATA) assistant. Generate a (question, answer) pair for RAG benchmarking based on a security blog snippet and MITRE ATT&CK framework reference. Use only the provided information. Do not invent facts.

- - - - - - - - - - - - - - - - - - - - - - - - - - - - - - - - - - - - - - - - - - - - - - - - -

INPUTS
- Source blog (attack description): {SOURCE_NODE}
- MITRE ATT&CK framework reference: {TARGET_NODE}

- - - - - - - - - - - - - - - - - - - - - - - - - - - - - - - - - - - - - - - - - - - - - - - - -

GUIDELINES
- Formulate a concise Question (1–2 sentences). Begin with: I recently read in a security blog: "..." Then ask to identify the MITRE ATT&CK category and explain the mapping.
- Additionally, request the relevant aspect {ASPECT}:
  - If {ASPECT} = kill_chain_phases: ask where in the attack lifecycle the behavior occurs.
  - If {ASPECT} = x_mitre_detection: ask how it can be detected (logs, sensors, NDR).
  - If {ASPECT} = x_mitre_platforms: ask which operating systems are affected.
  - If {ASPECT} = x_mitre_data_sources: ask which telemetry/logs can detect it.
- Answer should use a numbered list to provide:
  - MITRE ATT&CK technique ID; Mapping explanation; Response to the selected {ASPECT}.

- - - - - - - - - - - - - - - - - - - - - - - - - - - - - - - - - - - - - - - - - - - - - - - - -

EXAMPLE
Question: The attack involves crafting Microsoft Office documents to circumvent the Mark of the Web security feature. Please provide:

- The MITRE ATT&CK technique that maps to this behavior.
- An explanation of the mapping.
- The kill chain phase.

Answer:

- T1204.002 – User Execution: Malicious File.
- Mapping Explanation: Crafting Office documents to bypass security features aligns with T1204.002, involving tricking users into executing malicious files.
- Kill Chain Phase: Execution.

- - - - - - - - - - - - - - - - - - - - - - - - - - - - - - - - - - - - - - - - - - - - - - - - -

OUTPUT
Respond with only the JSON object:

{ "question": "Your question here"; "answer": "Your answer with bullet points" }

---

Vulnerability Classification and Attribution (VCA) belongs to the Dynamic CTI tasks, where the model aligns security blog vulnerability descriptions with the CWE framework. In this prompt, variable transformation follows the same procedure as in ATA.

---

**Prompt for VCA Generation**

INSTRUCTION

You are a Vulnerability Classification and Attribution (VCA) assistant. Generate a (question, answer) pair for RAG benchmarking based on a security blog snippet and CWE framework reference. Use only the provided information. Do not invent facts.

---

INPUTS

- Source blog (vulnerability description): {SOURCE_NODE}
- CWE framework reference: {TARGET_NODE}

---

GUIDELINES

- Formulate a concise Question (1–2 sentences). Begin with: I recently read in a security blog: "..." Then ask to identify the CWE category and explain the mapping.
- Additionally, request the relevant aspect {ASPECT}:
    - If {ASPECT} = Applicable_Platforms: ask what platforms or stacks are impacted.
    - If {ASPECT} = Modes_Of_Introduction: ask at what stage the issue is introduced.
    - If {ASPECT} = Likelihood_Of_Exploit: ask how often attackers exploit it in real-world attacks.
    - If {ASPECT} = Common_Consequences: ask what consequences or risks it causes.
    - If {ASPECT} = Detection_Methods: ask how it can be detected.
    - If {ASPECT} = Potential_Mitigations: ask what fixes (code or config changes) are recommended.
- Answer should use a numbered list to provide:
    - CWE ID.
    - Mapping explanation.
    - Response to the selected aspect {ASPECT}.

---

EXAMPLE

Question: I recently read in a security blog: "A web application fails to properly validate file uploads, allowing attackers to upload malicious executable files that can be executed on the server." Please provide:

- The CWE category that maps to this vulnerability.
- An explanation of the mapping.
- The potential mitigations.

Answer:

- CWE-434 – Unrestricted Upload of File with Dangerous Type.
- Mapping Explanation: Failure to validate file types/extensions before allowing uploads aligns directly with CWE-434.
- Potential Mitigations:
    - Implement strict allowlist validation of file types.
    - Verify content-type beyond file extensions.
    - Store uploaded files outside the web root.
    - Scan files for malicious content before storage.
    - Use secure upload libraries with built-in validation.

---

OUTPUT

Respond with only the JSON object:

```
{
    "question": "Your question here",
    "answer": "Your answer with bullet points"
}
```

Root Cause Mapping (RCM) belongs to the Structured CTI tasks, where the model aligns CVE entries with CWE references to identify root causes of vulnerabilities. In this prompt, variables enclosed in curly braces (e.g., {SOURCE_NODE}, {TARGET_NODE}) are placeholders dynamically replaced at runtime. Specifically, {SOURCE_NODE} provides the CVE input, while {TARGET_NODE} supplies the CWE reference for mapping.

---

**Prompt for RCM Generation**

INSTRUCTION

You are a Cyber Threat Intelligence (CTI) assistant. Given one CVE entry (JSON) as the source and one CWE entry (JSON) as the target, generate a (question, answer) pair for benchmarking RAG on the Root Cause Mapping (RCM) task. Use only the provided inputs. Do not invent facts.

- - - - - - - - - - - - - - - - - - - - - - - - - - - - - - - - -

INPUTS

- SOURCE CVE: {SOURCE_NODE}
- TARGET CWE: {TARGET_NODE}

- - - - - - - - - - - - - - - - - - - - - - - - - - - - - - - - -

GUIDELINES

- Formulate a Question that:
    - References the source CVE (by ID or title).
    - Asks which CWE it maps to.
    - Asks why this CWE is the root cause.
    - Avoids explicitly naming the CWE ID in the question.
- Answer should be a bullet list containing:
    - CWE ID (e.g., "CWE-770").
    - Explicit reference to both CVE and CWE with short mapping explanation.
    - Each bullet <= 22 words.

- - - - - - - - - - - - - - - - - - - - - - - - - - - - - - - - -

EXAMPLE

Question: Which CWE is the root cause of CVE-2018-6869? Explain why that CWE is the root cause of CVE-2018-6869?

Answer:

- CWE-770.
- CVE-2018-6869 causes uncontrolled memory allocation, matching CWE-770: allocation of resources without limits.

- - - - - - - - - - - - - - - - - - - - - - - - - - - - - - - - -

OUTPUT

Respond with only the JSON object:

```
{
    "question": "Your question here",
    "answer": "− CWE−XXX.\n− CVE−XXXX causes ..., matching CWE−XXX: ..."
}
```

Weakness Instantiation Mapping (WIM) belongs to the Structured CTI tasks, where the model aligns CWE entries with CVE instantiations to determine how specific vulnerabilities embody generalized weaknesses. In this prompt, variable transformation follows the same procedure as in RCM.

---

**Prompt for WIM Generation**

INSTRUCTION
You are a Cyber Threat Intelligence (CTI) assistant. Given one CWE entry (JSON) as the source and one CVE entry (JSON) as the target, generate a (question, answer) pair for benchmarking RAG on the Weakness Instantiation Mapping (WIM) task. Use only the provided inputs. Do not invent facts.

- - - - - - - - - - - - - - - - - - - - - - - - - - - - - - - - - - - - - - - -

INPUTS
- SOURCE CWE: {SOURCE_NODE}
- TARGET CVE: {TARGET_NODE}

- - - - - - - - - - - - - - - - - - - - - - - - - - - - - - - - - - - - - - - -

GUIDELINES
- Formulate a Question that:
    - References the source CWE (by ID or title).
    - Asks which CVE instantiates this weakness.
    - Asks why this CVE demonstrates the weakness.
    - Avoids explicitly naming the CVE ID in the question.
- Answer should be a bullet list containing:
    - CVE ID (e.g., "CVE-2018-6869").
    - Explicit reference to both CWE and CVE with short instantiation explanation.
    - Each bullet <= 22 words.

- - - - - - - - - - - - - - - - - - - - - - - - - - - - - - - - - - - - - - - -

EXAMPLE
Question: Which CVE instantiates CWE-770? Explain the reason why that CVE demonstrates CWE-770?

Answer:

- CVE-2018-6869.
- CVE-2018-6869 demonstrates uncontrolled memory allocation, instantiating CWE-770: allocation of resources without limits.

- - - - - - - - - - - - - - - - - - - - - - - - - - - - - - - - - - - - - - - -

OUTPUT
Respond with only the JSON object:

```
{
    "question": "Your question here",
    "answer": "– CVE–XXXX.\n– CVE–XXXX demonstrates ..., instantiating CWE–XXX: ..."
}
```

Attack Technique Derivation (ATD) belongs to the Structured CTI tasks, where the model links CAPEC attack patterns to MITRE ATT&CK techniques, thereby deriving atomic adversarial behaviors. In this prompt, variable transformation follows the same procedure as in RCM.

---

**Prompt for ATD Generation**

INSTRUCTION
You are a Cyber Threat Intelligence (CTI) assistant. Given one CAPEC entry (JSON) as the source and one MITRE ATT&CK entry (JSON) as the target, generate a (question, answer) pair for benchmarking RAG on the Attack Technique Derivation (ATD) task. Use only the provided inputs. Do not invent facts.

INPUTS

- SOURCE CAPEC: {SOURCE_NODE}
- TARGET MITRE ATT&CK: {TARGET_NODE}

GUIDELINES

- Formulate a Question that:
  - References the source CAPEC (by ID or title).
  - Asks which MITRE ATT&CK technique maps to this attack pattern.
  - Asks why this technique corresponds to the pattern.
  - Avoids explicitly naming the MITRE ATT&CK technique ID in the question.
- Answer should be a bullet list containing:
  - MITRE ATT&CK technique ID (e.g., "T1059.001").
  - Explicit reference to both CAPEC and ATT&CK with a short mapping explanation.
  - Each bullet <= 22 words.

EXAMPLE
Question: Which MITRE ATT&CK technique maps to CAPEC-242? Explain the reason why that technique corresponds to CAPEC-242?

Answer:

- T1059.001.
- T1059.001 corresponds to PowerShell command execution, mapping CAPEC-242: using PowerShell for command execution.

OUTPUT
Respond with only the JSON object:

```
{
    "question": "Your question here",
    "answer": "– TXXXX.\n– TXXXX corresponds to ..., mapping CAPEC–XXX: ..."
}
```

---

Exploitation Surface Discovery (ESD) belongs to the Structured CTI tasks, where the model links CWE weaknesses to CAPEC attack patterns, thereby identifying potential exploitation paths. In this prompt, variable transformation follows the same procedure as in RCM.

---

**Prompt for ESD Generation**

INSTRUCTION
You are a Cyber Threat Intelligence (CTI) assistant. Given one CWE entry (JSON) as the source and one CAPEC entry (JSON) as the target, generate a (question, answer) pair for benchmarking RAG on the Exploitation Surface Discovery (ESD) task. Use only the provided inputs. Do not invent facts.

INPUTS

- SOURCE CWE: {SOURCE_NODE}
- TARGET CAPEC: {TARGET_NODE}

GUIDELINES

- Formulate a Question that:
  - References the source CWE (by ID or title).
  - Asks which CAPEC attack pattern exploits this weakness.
  - Asks why the CAPEC pattern can exploit the CWE vulnerability.
  - Avoids explicitly naming the CAPEC ID in the question.
- Answer should be a bullet list containing:
  - CAPEC attack pattern ID (e.g., "CAPEC-63").
  - Explicit reference to both CAPEC and CWE with a short mapping explanation.
  - Each bullet <= 22 words.

EXAMPLE
Question: Which CAPEC attack pattern exploits CWE-79? Explain the reason why that CAPEC pattern can exploit CWE-79?

Answer:

- CAPEC-63.
- CAPEC-63 exploits cross-site scripting vulnerability, targeting CWE-79: improper neutralization of input during web page generation.

OUTPUT
Respond with only the JSON object:

```
{
  "question": "Your question here",
  "answer": "− CAPEC−XXX.\n− CAPEC−XXX exploits ..., targeting CWE−XXX: ..."
}
```

Campaign Storyline Construction (CSC) belongs to the Unstructured CTI tasks, where the model integrates multiple heterogeneous reports to extract campaign-level threat intelligence. Variables enclosed in curly braces (e.g., {SOURCE_NODE}, {TARGET_NODES}, {ASPECT}) are placeholders dynamically replaced at runtime. Specifically, {SOURCE_NODE} provides the seed report, {TARGET_NODES} includes related contextual reports, and {ASPECT} controls which intelligence attribute (entities, dates, targeting, tools) is queried.

---

**Prompt for CSC Generation**

INSTRUCTION
You are a Campaign Storyline Construction (CSC) assistant. Generate a (question, answer) pair for RAG benchmarking based on security reports to extract campaign intelligence. Use only the provided information. Do not invent facts.

INPUTS
- Source blog: {SOURCE_NODE}
- Related blogs: {TARGET_NODES}

GUIDELINES
- Formulate a Question (1–2 sentences) that requests campaign-level intelligence from cross-report analysis.
- Depending on {ASPECT}, the question should ask for:
    - If {ASPECT} = entities: ask for common entities (malware, threat actors, CVEs).
    - If {ASPECT} = dates: ask for campaign time span (earliest and latest reports).
    - If {ASPECT} = targeting: ask for primary target industries/regions.
    - If {ASPECT} = tool: ask for canonical tools or malware consistently used.
- Answer should:
    - Provide extracted campaign intelligence.
    - Address the selected aspect {ASPECT}.
    - Avoid referring to blog IDs.
    - Use bullet points for clarity.

EXAMPLE
Question: What malware families, threat groups, or vulnerability IDs appear consistently across these blogs?

Answer:

- Malware Families: Emotet, TrickBot
- Threat Actor Groups: APT28, Lazarus Group
- Vulnerability IDs: CVE-2021-34527, CVE-2020-1472

OUTPUT
Respond with only the JSON object:

```
{
    "question": "Your question here",
    "answer": "Your answer with bullet points"
}
```

Threat Actor Profiling (TAP) belongs to the Unstructured CTI tasks, where the model integrates multiple heterogeneous reports to extract threat actor intelligence. In this prompt, variable transformation follows the same procedure as in CSC.

---

**Prompt for TAP Generation**

INSTRUCTION

You are a Threat Actor Profiling (TAP) assistant. Generate a (question, answer) pair for RAG benchmarking based on security reports to extract threat actor intelligence. Use only the provided information. Do not invent facts.

- - - - - - - - - - - - - - - - - - - - - - - - - - - - - - - - - - - - - - - -

INPUTS

- Source blog: {SOURCE_NODE}
- Related blogs: {TARGET_NODES}

- - - - - - - - - - - - - - - - - - - - - - - - - - - - - - - - - - - - - - - -

GUIDELINES

- Formulate a Question (1–2 sentences) that requests threat actor intelligence from cross-report analysis.
- Depending on {ASPECT}, the question should ask for:
  - If {ASPECT} = actor: ask for canonical threat actor name and alias resolution.
  - If {ASPECT} = tool: ask for primary tool or malware consistently attributed.
  - If {ASPECT} = target: ask for primary target sectors or regions.
- Answer should:
  - Provide extracted threat actor intelligence.
  - Address the selected aspect {ASPECT}.
  - Avoid referring to blog IDs.
  - Use bullet points for clarity.

- - - - - - - - - - - - - - - - - - - - - - - - - - - - - - - - - - - - - - - -

EXAMPLE

Question: What is the canonical threat actor name, resolving any aliases mentioned across these reports?

Answer:

- Canonical Threat Actor Name: APT29
- Aliases Resolved: Cozy Bear, Nobelium
- Primary Tool/Malware: Cobalt Strike, Mimikatz

- - - - - - - - - - - - - - - - - - - - - - - - - - - - - - - - - - - - - - - -

OUTPUT

Respond with only the JSON object:

```
{
   "question": "Your question here",
   "answer": "Your answer with bullet points"
}
```

---

Malware Lineage Analysis (MLA) belongs to the Unstructured CTI tasks, where the model integrates multiple heterogeneous reports to extract malware lineage intelligence. In this prompt, variable transformation follows the same procedure as in CSC.

---

**Prompt for MLA Generation**

INSTRUCTION
You are a Malware Lineage Analysis (MLA) assistant. Generate a (question, answer) pair for RAG benchmarking based on security reports to extract malware lineage intelligence. Use only the provided information. Do not invent facts.

- - - - - - - - - -

INPUTS

- Source blog: {SOURCE_NODE}
- Related blogs: {TARGET_NODES}

- - - - - - - - - -

GUIDELINES

- Formulate a Question (1–2 sentences) that requests malware lineage intelligence from cross-report analysis.
- Depending on {ASPECT}, the question should ask for:
  - If {ASPECT} = variant: ask for all distinct malware variants across reports.
  - If {ASPECT} = capability: ask for the most notable new capability added in later variants.
- Answer should:
  - Provide extracted malware lineage intelligence.
  - Address the selected aspect {ASPECT}.
  - Avoid referring to blog IDs.
  - Use bullet points for clarity.

- - - - - - - - - -

EXAMPLE
Question: What are all the distinct malware variant names mentioned across these reports?

Answer:

- Emotet v1.0
- Emotet v2.0
- Emotet v3.0
- Emotet v4.0
- Most Notable New Capability: Added worm-like propagation in v3.0

- - - - - - - - - -

OUTPUT
Respond with only the JSON object:

```
{
   "question": "Your question here",
   "answer": "Your answer with bullet points"
}
```

---

## E.2  LLM-AS-A-JUDGE

We adopt an LLM-as-a-Judge framework to triage QA pairs generated for CTI analysis. The judge model applies a structured rubric that evaluates each sample along five dimensions: (i) factual correctness and grounding in the provided CTI evidence, (ii) relevance to concrete security entities or activities, (iii) clarity and unambiguity of the language, (iv) consistency with the question intent, and (v) completeness of reasoning. Responses containing hallucinations or unverifiable claims receive lower scores.

*Filtering policy*: The judge returns a single holistic score in [1,5]. We retain only items with a score >= 3 for expert second-stage review; items with scores < 3 are discarded to reduce reviewer burden and improve data reliability. This design ensures (1) reproducible screening with transparent criteria, (2) high recall of plausibly-correct items for expert adjudication, and (3) a clean separation between automatic triage and human confirmation.

---

**Prompt for LLM-as-a-Judge**

INSTRUCTION
You are a cybersecurity expert and a rigorous evaluator of question–answer (QA) pairs in the cyber threat intelligence (CTI) domain. Judge each sample using the rubric below and produce a single holistic score from 1 to 5.

- - - - - - - - - - - - - - - - - - - - - - - - - - - - - - - - - - - - -

INPUTS
CTI Document(s): {document_text}
Question: {question}
Answer: {answer}

- - - - - - - - - - - - - - - - - - - - - - - - - - - - - - - - - - - - -

EVALUATION RUBRIC

1. **Factual Correctness & Grounding** — Is the answer factually correct and directly supported by the provided CTI document (if any)? Avoid hallucinations and unverifiable claims.
2. **Relevance** — Does the QA focus on concrete cybersecurity entities (exposures, techniques, assets, indicators) rather than trivial or generic content?
3. **Clarity** — Is the question unambiguous and is the answer concise and machine-usable in CTI systems?
4. **Task Consistency** — Does the answer faithfully address the asked question and remain logically consistent?
5. **Completeness** — Does the answer include the key technical context and reasoning needed for CTI usage?

**Scoring Scale (1–5)**

- **5** — Fully correct, well-grounded, clear, task-faithful, and complete.
- **4** — Correct and grounded with minor omissions or minor stylistic issues.
- **3** — Largely plausible but with partial grounding or limited completeness; acceptable for expert re-check.
- **2** — Weak grounding, notable gaps or ambiguity; likely unusable without major fixes.
- **1** — Incorrect or hallucinated; off-task or unverifiable.

- - - - - - - - - - - - - - - - - - - - - - - - - - - - - - - - - - - - -

OUTPUT
Return only the following JSON object (no extra text or markdown):

```
{
    "assessment": "<concise justification>",
    "score": "<integer 1–5>"
}
```

---

### E.3 Prompt in Dataset Evaluation

Root Cause Mapping (RCM) evaluates whether models can correctly associate a vulnerability instance with its underlying software weakness. In particular, the task requires identifying the CWE that serves as the fundamental cause of a given CVE.

Variables enclosed in curly braces (e.g., {CVE_ID}, {entity_id}, {description}, {related_cwes}, {cwe_detail}) represent placeholders dynamically substituted at runtime. Specifically, {CVE_ID} denotes the vulnerability under evaluation, {entity_id} captures the case context, while {description} and {related_cwes} provide supporting metadata. The {cwe_detail} field enumerates candidate CWE entries along with their textual descriptions.

---

**Prompt for RCM Evaluation**

INSTRUCTION
You are a Root Cause Mapping (RCM) expert. Answer questions about cybersecurity mappings based on your knowledge. For Root Cause Mapping, provide the CWE ID that represents the underlying cause of the given CVE, along with a concise explanation.

- - - - - - - - - - - - - - - - - - - - - - - - - - - - - - - - -

INPUTS
Based on your knowledge, which CWE does {CVE_ID} map to? Please provide the CWE ID and a brief explanation.

USER_PROMPT_WITH_DESCRIPTION: Based on the following information about {entity_id}, which CWE does it map to?

- Description: {description}
- Related CWEs: {related_cwes}

CWE Details: - {cwe_detail.id}: {cwe_detail.description[:200]} - ...

- - - - - - - - - - - - - - - - - - - - - - - - - - - - - - - - -

OUTPUT
Please provide the CWE ID and a brief explanation.

---

Weakness Instantiation Mapping (WIM) evaluates whether models can correctly associate an abstract software weakness with its concrete vulnerability instances. Specifically, the task requires identifying the CVE that instantiates a given CWE. In this prompt, variable transformation follows the same procedure as in RCM.

---

**Prompt for WIM Evaluation**

INSTRUCTION
You are a Weakness Instantiation Mapping (WIM) expert. Answer questions about cybersecurity mappings based on your knowledge. For Weakness Instantiation Mapping, identify the CVE that instantiates the given CWE and provide a concise explanation.

- - - - - - - - - - - - - - - - - - - - - - - - - - - - - - - - -

INPUTS
Based on your knowledge, which CVE instantiates {CWE_ID}? Please provide the CVE ID and a brief explanation.

USER_PROMPT_WITH_DESCRIPTION: Based on the following information about {entity_id}, which CVE instantiates it?

- Description: {description}
- Related CVEs: {related_cves}

CVE Details: - {cve_detail.id}: {cve_detail.description[:200]} - ...

- - - - - - - - - - - - - - - - - - - - - - - - - - - - - - - - -

OUTPUT
Please provide the CVE ID and a brief explanation.

---

Attack Technique Derivation (ATD) task evaluates whether models can correctly associate abstract attack patterns with corresponding adversarial techniques. ATD requires identifying the MITRE ATT&CK technique that best maps to a given CAPEC entry. Variables enclosed in curly braces (e.g., {CAPEC_ID}, {entity_id}, {description}, {related_mitre}, {mitre_detail}) represent placeholders dynamically substituted at runtime. Specifically, {CAPEC_ID} denotes the attack pattern under evaluation, {entity_id} captures the case context, while {description} and {related_mitre} provide supporting metadata. The {mitre_detail} field enumerates candidate ATT&CK techniques along with their textual descriptions.

---

**Prompt for ATD Evaluation**

INSTRUCTION
You are an Attack Technique Derivation (ATD) expert. Answer questions about cybersecurity mappings based on your knowledge. For ATD, provide the MITRE ATT&CK technique ID that corresponds to the given CAPEC attack pattern, along with a concise explanation.

- - - - - - - -

INPUTS
Based on your knowledge, which MITRE ATT&CK technique maps to {CAPEC_ID}? Please provide the technique ID and a brief explanation.

USER_PROMPT_WITH_DESCRIPTION: Based on the following information about {entity_id}, which MITRE ATT&CK technique maps to it?

- Description: {description}
- Related MITRE ATT&CK techniques: {related_mitre}

MITRE ATT&CK Details: - {mitre_detail.id}: {mitre_detail.description[:200]} - ...

- - - - - - - -

OUTPUT
Please provide the MITRE ATT&CK technique ID and a brief explanation.

---

Exploitation Surface Discovery (ESD) task evaluates whether models can correctly identify attack patterns that exploit a specified software weakness. Specifically, the task requires selecting the CAPEC entry that best describes how adversaries can leverage the given CWE. In this prompt, variable transformation follows the same procedure as in RCM.

---

**Prompt for ESD Evaluation**

INSTRUCTION
You are an Exploitation Surface Discovery (ESD) expert. Answer questions about cybersecurity mappings based on your knowledge. For Exploitation Surface Discovery, identify the CAPEC attack pattern that can exploit the given CWE and provide a concise explanation.

- - - - - - - -

INPUTS
Based on your knowledge, which CAPEC attack pattern exploits {CWE_ID}? Please provide the CAPEC ID and a brief explanation.

USER_PROMPT_WITH_DESCRIPTION: Based on the following information about {entity_id}, which CAPEC attack pattern exploits it?

- Description: {description}
- Related CAPEC patterns: {related_capecs}

CAPEC Details: - {capec_detail.id}: {capec_detail.description[:200]} - ...

- - - - - - - -

OUTPUT
Please provide the CAPEC ID and a brief explanation.

---

Campaign Storyline Construction (CSC) task, evaluates the model's ability to synthesize campaign-level intelligence from heterogeneous security reports. Variables such as {query}, {context}, {clustering_context}, and {task_description} serve as dynamic placeholders: {query} denotes the latest report, {context} and {clustering_context} provide historical continuity, and {task_description} specifies the intelligence focus.

**Prompt for CSC Evaluation**

INSTRUCTION
You are a Campaign Storyline Construction (CSC) assistant. Analyze security reports to extract campaign intelligence. Use only the provided information. Do not invent facts.

INPUTS
Latest Blog: {query}
Related Historical Blogs: {context}{clustering_context}

QUESTION
- {task_description}
- Do not refer to the blogs by their IDs in the answer.

EXAMPLE
Based on the question: "What is the time span of the EleKtra-Leak campaign based on the earliest and latest report dates?"

Expected answer format:

- Earliest report date: December 2020
- Latest report date: October 6, 2023

OUTPUT
Please provide your answer:

Threat Actor Profiling (TAP) task evaluates the model's ability to consolidate threat actor intelligence from heterogeneous security reports. In this prompt, variable transformation follows the same procedure as in CSC.

**Prompt for TAP Evaluation**

INSTRUCTION
You are a Threat Actor Profiling (TAP) assistant. Analyze security reports to extract threat actor intelligence.
Use only the provided information. Do not invent facts.

INPUTS
Latest Blog: {query}
Related Historical Blogs: {context}{clustering_context}

QUESTION
- {task_description}
- Do not refer to the blogs by their IDs in the answer.

EXAMPLE
Based on the question: "What is the canonical threat actor name, resolving any aliases mentioned across these reports?"

Expected answer format:

- Canonical Threat Actor Name: APT29
- Aliases Resolved: Cozy Bear, Nobelium
- Primary Tool/Malware: Cobalt Strike, Mimikatz

OUTPUT
Please provide your answer:

Malware Lineage Analysis (MLA) task evaluates the model's ability to reconstruct malware evolution from heterogeneous security reports. In this prompt, variable transformation follows the same procedure as in CSC.

---

**Prompt for MLA Evaluation**

INSTRUCTION
You are a Malware Lineage Analysis (MLA) assistant. Analyze security reports to extract malware lineage intelligence.
Use only the provided information. Do not invent facts.

- - - - - - - - - - - - - - - - - - - - - - - - - - - - - - - - - - - - - - - - - - -

INPUT
Latest Blog: {query}
Related Historical Blogs: {context}{clustering_context}

- - - - - - - - - - - - - - - - - - - - - - - - - - - - - - - - - - - - - - - - - - -

QUESTION

- {task_description}
- Do not refer to the blogs by their IDs in the answer.

- - - - - - - - - - - - - - - - - - - - - - - - - - - - - - - - - - - - - - - - - - -

EXAMPLE
Based on the question: "What are all the distinct malware variant names mentioned across these reports?"

Expected answer format:

- Emotet v1.0
- Emotet v2.0
- Emotet v3.0
- Emotet v4.0
- Most Notable New Capability: Added worm-like propagation in v3.0

- - - - - - - - - - - - - - - - - - - - - - - - - - - - - - - - - - - - - - - - - - -

OUTPUT
Please provide your answer:

---

Closed-Book Attack Technique Attribution (ATA) task evaluates whether models can align textual attack descriptions with the MITRE ATT&CK framework without external evidence. Variables such as {question} represent dynamically substituted inputs, where the model must output (i) the relevant technique ID, (ii) a clear mapping rationale, and (iii) corresponding detection recommendations aligned to the {x_mitre_detection} field.

---

**Prompt for Closed-Book Evaluation of ATA**

INSTRUCTION
You are an Attack Technique Attribution (ATA) expert. Based on the following security question, please provide:

- The MITRE ATT&CK technique ID (e.g., T1059.001 – PowerShell)
- An explanation of how the behavior maps to that technique
- Detection recommendations aligned to x_mitre_detection (logs, sensors, NDR)

- - - - - - - - - - - - - - - - - - - - - - - - - - - - - - - - - - - - - - - - - - -

INPUT
Question: {question}

- - - - - - - - - - - - - - - - - - - - - - - - - - - - - - - - - - - - - - - - - - -

OUTPUT
Please format your response as:

- 1) T####(.###) – [Technique/Sub-technique Name]
- 2) [Detailed mapping explanation]
- 3) [Detection guidance]

---

Close-book Vulnerability Classification and Attribution (VCA) task evaluates a model's ability to map a natural-language vulnerability description to the appropriate CWE category without external evidence. Given the input {question}, the model must (i) output the canonical CWE label (e.g., "CWE-434: Unrestricted Upload of File with Dangerous Type"), (ii) justify the mapping by linking salient behaviors in the description to the CWE definition, and (iii) recommend actionable mitigations (code or configuration). This close-book setting isolates classification and reasoning quality by removing retrieval effects, thereby assessing generalization from the standardized CWE taxonomy to previously unseen textual descriptions.

---

**Prompt for Closed-Book Evaluation of VCA**

INSTRUCTION
You are a Vulnerability Classification and Attribution (VCA) expert. Based on the following security question, please provide:

- The CWE category that maps to this vulnerability
- An explanation of how this vulnerability maps to the CWE
- What are the potential mitigations (code or configuration changes) to address this issue

- - - - - - - - - - - - - - - - - - - - - - - - - - - - - - - - - - - - - - - - -

INPUT
Question: {question}

- - - - - - - - - - - - - - - - - - - - - - - - - - - - - - - - - - - - - - - - -

OUTPUT
Please format your response as:

- 1) CWE-XXX: [CWE Name]
- 2) [Detailed explanation of the mapping]
- 3) [List of potential mitigations]

---

Vanilla Retrieval-Augmented Generation (RAG) setting for Attack Technique Attribution (ATA) assesses whether models can correctly align adversarial behaviors with MITRE ATT&CK techniques when provided with retrieved evidence. Given the input {question} and {evidence}, the model is required to (i) identify the appropriate ATT&CK technique ID, (ii) explain how the described behavior corresponds to that technique, and (iii) provide detection guidance aligned with {x_mitre_detection}. This setup isolates the model's ability to ground its reasoning in retrieved ATT&CK knowledge, thereby measuring retrieval utility and attribution precision under evidence-supported conditions.

---

**Prompt for Vanilla Evaluation of ATA**

INSTRUCTION
You are an Attack Technique Attribution (ATA) expert. Based on the following security question and retrieved ATT&CK evidence, please provide:

- The MITRE ATT&CK technique ID (e.g., T1059.001 – PowerShell)
- An explanation of how the behavior maps to that technique
- Detection recommendations aligned to x_mitre_detection (logs, sensors, NDR)

- - - - - - - - - - - - - - - - - - - - - - - - - - - - - - - - - - - - - - - - -

INPUTS
Question: {question}
Retrieved ATT&CK Evidence: {evidence}

- - - - - - - - - - - - - - - - - - - - - - - - - - - - - - - - - - - - - - - - -

OUTPUT
Please format your response as:

- 1) T####(.###) – [Technique/Sub-technique Name]
- 2) [Detailed mapping explanation]
- 3) [Detection guidance]

---

Vanilla RAG setting for VCA evaluates whether models can correctly associate vulnerabilities with CWE categories when provided with retrieved evidence. In this prompt, variable transformation follows the same procedure as in ATA.

---

**Prompt for Vanilla Evaluation of VCA**

INSTRUCTION

You are a cybersecurity expert. Based on the following security question and retrieved CWE evidence, please provide:

- The CWE category that maps to this vulnerability
- An explanation of how this vulnerability maps to the CWE
- What are the potential mitigations (code or configuration changes) to address this issue

- - - - - - - - - - - - - - - - - - - - - - - - - - - - - - - - - - - - - - - -

INPUTS

Question: {question}
Retrieved CWE Evidence: {evidence}

- - - - - - - - - - - - - - - - - - - - - - - - - - - - - - - - - - - - - - - -

OUTPUT

Please format your response as:

- 1) CWE-XXX: [CWE Name]
- 2) [Detailed explanation of the mapping]
- 3) [List of potential mitigations]

---

RAG Expansion setting for ATA and VCA tasks evaluates whether models can derive more accurate mappings by leveraging decomposed atomic behaviors. Variables enclosed in curly braces denote dynamic placeholders: {behavior_text} denotes the raw description of the observed adversarial behavior; {atomic_behaviors} lists the finer-grained actions derived from behavior decomposition; and {evidence} contains the selected ATT&CK or CWE matched against these atomic units.

For ATA, this means identifying the correct MITRE ATT&CK technique; for VCA, mapping to the most suitable CWE.

---

**Prompt for RAG Expansion Evaluation in ATA/VCA**

INSTRUCTION

You are a cybersecurity expert. Based on the following security question and the selected ATT&CK evidence from atomic behavior analysis, please provide a confident answer.

- - - - - - - - - - - - - - - - - - - - - - - - - - - - - - - - - - - - - - - -

INPUTS

Question: {question}
Original Behavior: {behavior_text}
Decomposed Atomic Behaviors: {atomic_behaviors}
Selected ATT&CK / CWE Evidence (from atomic behavior matching): {evidence}

- - - - - - - - - - - - - - - - - - - - - - - - - - - - - - - - - - - - - - - -

GUIDELINES

- You can answer based on the provided ATT&CK / CWE evidence, or use your internal knowledge if you're confident
- You don't have to select from the provided CWEs if you have a better answer
- Be confident in your response — if you're not sure, say so
- Provide the most accurate MITRE ATT&CK / CWE technique mapping

- - - - - - - - - - - - - - - - - - - - - - - - - - - - - - - - - - - - - - - -

OUTPUT

Please format your response as:

- 1) T####(.###) – [Technique/Sub-technique Name]
- 1) CWE-XXX: [CWE Name] (or "CWE-Unknown" if uncertain)
- 2) [Detailed explanation of the mapping and confidence level]
- 3) [Detection guidance aligned to x_mitre_detection]
- 3) [List of potential mitigations]

---

In hybrid evaluation settings, the pipeline first attempts to ground responses in external knowledge sources. When no relevant evidence can be retrieved through atomic behavior matching, the model is instructed to rely on its internal knowledge to produce an answer. This prompt thereby serves as

a fallback mechanism, ensuring that the model continues to generate useful intelligence even under evidence-sparse conditions.

> **Prompt for Internal Knowledge in ATA/VCA (No CWE/MITRE Evidence)**
>
> No relevant CWE/MITRE evidence was found through atomic behavior matching. Please use your internal knowledge to provide the best possible answer.

The Judge-CWE-Mapping prompt is employed in the VCA task to validate whether a given atomic behavior corresponds to a candidate CWE entry.

Variables enclosed in curly braces act as dynamic placeholders: {atomic_behavior} represents the extracted fine-grained behavior under analysis; {full_vulnerability} provides broader context for the CVE instance; {label} indicates the class of candidate mappings (e.g., CWE entries); {entry_label}, {candidate.title}, and {candidate.description} provide textual details of each candidate; {candidate.similarity_score} quantifies retrieval similarity.

By requiring binary ("YES"/"NO") judgments in JSON array format, this prompt enforces precise, auditable validation of candidate mappings, supporting robust evaluation of model alignment with CWE classification.

> **Prompt for Judge-CWE-Mapping in VCA**
>
> INSTRUCTION
> You are a cybersecurity expert. Please analyze whether the following atomic behavior has a mapping relationship with the given {label} entries.
>
> ---
>
> INPUTS
> Atomic Behavior: "{atomic_behavior}"
> Full Vulnerability Context: "{full_vulnerability}"
> {label} Candidates (with similarity scores):
> {i}. {entry_label}: {candidate.title}
> Description: {candidate.description[:500]}...
> Similarity Score: {candidate.similarity_score:.3f}
>
> ---
>
> GUIDELINES
> Please analyze each candidate and determine if it has a mapping relationship with the atomic behavior. Consider:
>
> - Semantic similarity between the atomic behavior and CWE description
> - Whether the CWE describes the same type of weakness/vulnerability
> - The similarity score (higher scores indicate better matches)
>
> **IMPORTANT:**
>
> - You can select at most one candidate as the best match
> - If no candidates are suitable, select none
> - If multiple candidates are suitable, choose the one with the highest similarity score
>
> ---
>
> OUTPUT
> For each candidate, respond with:
>
> - "YES" if it is the BEST match (at most one)
> - "NO" for all others
>
> Format your response as a JSON array with the same number of elements as candidates, e.g.: ["YES", "NO", "NO", ...] or ["NO", "NO", "NO", ...] if no match.
>
> ---
>
> OUTPUT
> Return only the JSON array.

The prompt for Decompose Vulnerability is applied in both the ATA and VCA tasks to transform a textual vulnerability description into a structured set of atomic behaviors.

Variables in curly braces act as placeholders: {vulnerability_text} denotes the raw vulnerability description to be analyzed. The output enforces a JSON array of 2–4 atomic behaviors, ensuring the representation is both concise and actionable.

---

**Prompt for Decompose Vulnerability in ATA/VCA**

INSTRUCTION
You are a cybersecurity expert. Analyze the following vulnerability description and identify its core atomic behaviors.

- - - - - - - - - - - - - - - - - - - - - - - - - - - - - - - - - - - - - - - - - -

INPUT
Vulnerability: "{vulnerability_text}"

- - - - - - - - - - - - - - - - - - - - - - - - - - - - - - - - - - - - - - - - - -

GUIDELINES

- Extract 2–4 key atomic behaviors that describe the vulnerability itself
- Each behavior should be a specific, actionable security issue
- Focus on the vulnerability mechanics, not mitigations or solutions
- Use clear, concise language describing what the vulnerability does
- Avoid splitting into too many granular behaviors

- - - - - - - - - - - - - - - - - - - - - - - - - - - - - - - - - - - - - - - - - -

EXAMPLE FORMAT
["behavior 1", "behavior 2", "behavior 3"]

- - - - - - - - - - - - - - - - - - - - - - - - - - - - - - - - - - - - - - - - - -

OUTPUT
Return only the JSON array.

---

The Select-Best-CWE prompt is employed in the VCA task to evaluate whether models can identify the most appropriate software weakness for a given vulnerability.

Variables in curly braces indicate placeholders dynamically substituted at runtime: {vulnerability} provides the full vulnerability text, while {cwe_candidates} supply multiple candidate CWE entries, each annotated with {idx}, {c.cwe_id}, {c.title}, {c.description}, and a pre-computed {c.similarity_score}. The model should output a JSON array aligned with the number of candidates, assigning "YES" to the single best match and "NO" to all others.

---

**Prompt for Select-Best-CWE in VCA**

INSTRUCTION
You are a cybersecurity expert. Select which CWE best matches the COMPLETE vulnerability description below.

- - - - - - - - - - - - - - - - - - - - - - - - - - - - - - - - - - - - - - - - - -

INPUT
Vulnerability Description: "{vulnerability}"
CWE Candidates to compare:
{idx}. CWE-{c.cwe_id}: {c.title}
Description: {c.description[:500]}...
Similarity Score: {c.similarity_score:.3f}
...

- - - - - - - - - - - - - - - - - - - - - - - - - - - - - - - - - - - - - - - - - -

OUTPUT
Respond as a JSON array with exactly {len(cwe_candidates)} elements in order where only the best candidate is "YES": ["YES", "NO", ...]

- - - - - - - - - - - - - - - - - - - - - - - - - - - - - - - - - - - - - - - - - -

OUTPUT
Return only the JSON array.

---