# OpenReview forum: "CTIArena: Benchmarking LLM Knowledge and Reasoning across Heterogeneous Cyber Threat Intelligence"
_ICLR.cc/2026/Conference — Submitted to ICLR 2026_

### Official Review · Reviewer_oucB · 2025-10-29

**Soundness:** 1
**Presentation:** 2
**Contribution:** 1
**Rating:** 2
**Confidence:** 5

**Summary:**

The paper introduces CTIArena, a benchmark designed to evaluate LLMs on diverse CTI tasks under both closed-book and knowledge-augmented settings. It defines nine tasks across structured, unstructured, and hybrid categories: spanning mappings among CVE, CWE, CAPEC, and ATT&CK taxonomies. The authors evaluate ten state-of-the-art LLMs and demonstrate that while general-purpose models perform poorly in closed-book CTI reasoning, performance improves significantly with security-specific retrieval methods.

**Strengths:**

The paper offers limited strengths in terms of novelty, clarity, or significance. While it identifies several limitations of prior CTI benchmarks, such as CTIBench and SEvenLLM, including their narrow task coverage and lack of retrieval-augmented evaluation, it does not necessarily address these limitations. Although the authors aim to develop a more comprehensive benchmark for CTI reasoning, as discussed below, the resulting CTIArena framework still falls short of constituting a true “reasoning” benchmark for CTI contexts.

**Weaknesses:**

The main limitation of this work is that CTIArena is not truly a benchmark for evaluating LLM reasoning. The included tasks primarily measure an LLM’s ability to extract or retrieve known knowledge, rather than perform any form of reasoning or inference. Consequently, the large performance gains observed with retrieval-augmented generation (RAG) are unsurprising, as once supporting documents are provided, the tasks essentially become lookup problems. For instance, in the RCM (Root Cause Mapping) structured task, the question simply asks for the CWE ID corresponding to a given CVE ID, without including any contextual information or description. Such a task can be trivially solved via a direct database or Google search, and since many CVEs postdate the training cutoff of the evaluated LLMs, it is expected that they would fail without retrieval access. This design does not test reasoning ability, nor is an LLM the appropriate tool for it. In contrast, CTIBench defines RCM using CVE descriptions rather than IDs, requiring models to infer the underlying weakness (CWE) from textual evidence, thus addressing a practical and reasoning-oriented use case for analyzing newly discovered vulnerabilities. CTIArena completely overlooks this reasoning aspect in its task design.

Moreover, the paper makes several unsubstantiated claims, such as stating that CTIBench contains only 150 manually annotated queries, whereas in reality the dataset is substantially larger. For example, the RCM task alone includes around 1,000 questions, while CTIArena itself contains only 691 total questions. Moreover, the two hybrid tasks, CTI-VCA and CTI-ATA, are very similar in nature to the CTI-RCM and CTI-ATE tasks from CTIBench.

**Questions:**

I would recommend that the authors either revise the task designs to better capture and evaluate the reasoning abilities of LLMs in CTI contexts by incorporating tasks that require inference, abstraction, or contextual understanding, rather than factual lookup or reframe the benchmark’s stated goal to focus explicitly on assessing whether LLMs can retrieve and align CTI concepts from heterogeneous knowledge sources.

---

> ### Author Response · Authors · 2025-11-20
> **Thank you for your valuable feedbacks.**
>
> ### Q: The benchmark appears to test factual lookup rather than genuine reasoning.
>
> **A:** CTIArena explicitly evaluates multiple forms of CTI reasoning.
>
> **Structured tasks:** reasoning over CTI schemas and relations.
> - **Closed-book:** Given a CTI entity, the model must recall how it is related to other CTI entities (e.g., the associated CWE or CAPEC) and justify why this relation is valid under the CTI schema.
> - **Knowledge-augmented:** Given retrieved database entries, the model must read structured fields, verify relational constraints, and decide which identifier is correctly linked under the schema, not rely on memorized facts.
>
> **Hybrid tasks:** grounding free-form CTI narrative text to formal CTI entities.
> - **Closed-book:** Perform end-to-end semantic interpretation and determine which CTI entry best explains the narrative using only parametric knowledge.
> - **Knowledge-augmented:** Given retrieved (potentially noisy) CTI entries, identify relevant fields, compare them with the narrative description, and select the entry most consistent with the reported behavior.
>
> **Unstructured tasks:** cross-report analytic synthesis.
> Retrieve, compare, and integrate evidence across multiple CTI reports; isolate relevant spans; reconcile heterogeneous signals (indicators, procedures, campaigns); and produce a coherent analytic conclusion.
>
> CTIArena therefore covers the major reasoning forms in CTI workflows: **schema-level relational reasoning**, **semantic grounding from narrative text**, and **multi-report analytic synthesis**.
>
> ---
>
> ### Q: The RCM task seems trivial because it maps a CVE ID directly to its CWE root cause, whereas CTIBench uses CVE descriptions.
>
> **A:** This difference is intentional. CTIBench’s RCM formulation merges two distinct reasoning operations:
>
> 1. **Grounding a narrative CVE description to the correct CVE ID (NL→CVE).**
> 2. **Mapping the identified CVE to its CWE root cause (CVE→CWE).**
>
> These steps rely on different reasoning skills and fail for different reasons. When merged, one cannot determine whether an incorrect answer reflects misinterpretation of the narrative or misunderstanding of the CWE correlation.
>
> CTIArena separates these components for diagnostic clarity:
>
> - **Hybrid tasks** evaluate **NL→CVE** grounding and semantic alignment.
> - **RCM** focuses solely on the **CVE→CWE** structured mapping using authoritative CTI entities.
>
> This decomposition mirrors real CTI pipelines and allows us to precisely diagnose where failures occur, instead of mixing multiple reasoning regimes into a single task.
>
> ---
>
> ### Q: CTI-VCA and CTI-ATA seem similar to CTIBench’s CTI-RCM and CTI-ATE.
>
> **A:** The tasks differ in their reasoning focus.
>
> - **CTI-VCA** directly evaluates **NL→CWE** reasoning and does **not** involve CVE identification.
>   - CTIBench’s CTI-RCM, in contrast, mixes NL→CVE grounding with CVE→CWE mapping.
>   - They therefore probe fundamentally different reasoning capabilities.
>
> - **CTI-ATA** and CTIBench’s **CTI-ATE** appear similar because ATT&CK attribution is a fundamental CTI analytic function that any comprehensive benchmark must include. Their overlap reflects the domain, not a benchmark artifact.
>
> CTIArena extends beyond prior benchmarks by defining a broader and more systematic task system across heterogeneous CTI sources and across both closed-book and knowledge-augmented settings.
>
> ---
>
> ### Q: The paper contains an incorrect statement about CTIBench’s size, while CTIArena has only 691 instances.
>
> **A:** We thank the reviewer for pointing out the scale error; this was a typo and will be corrected. The correction does not affect our main claims, which concern fundamental differences between CTIArena and prior CTI benchmarks: existing benchmarks cover narrow task families, use only single-source CTI, and evaluate exclusively in closed-book settings that do not reflect practical CTI workflows.
>
> Regarding dataset size, CTIArena was expanded during the rebuttal period to **1,433 instances**:
>
> - **303** unstructured
> - **641** structured
> - **489** hybrid (including two new tasks: **ECA** for NL→CAPEC and **ACA** for NL→CVE)
>
> We re-evaluated frontier closed-source models and lightweight open-source baselines; larger open-source models are still running due to resource constraints. **Model rankings and relative performance gaps remain fully consistent** with the original findings.
>
> For the new hybrid tasks, the results follow established hybrid patterns:
>
> - **ECA:** GPT-5 achieves **0.784 (closed-book)** and **0.865 (RAG)**; other models cluster around 0.4–0.5.
> - **ACA:** Overall harder. GPT-5 reaches **0.401** (others ≤0.28); **RAG improves GPT-5 to 0.692**.
>   - Difficulty arises from semantic mismatch: CVE entries use templated phrasing, whereas CTI reports use scenario-driven descriptions with limited lexical overlap.
>
> The expanded dataset and updated results will be incorporated into the revised manuscript.

---

### Official Review · Reviewer_qfcD · 2025-11-01

**Soundness:** 2
**Presentation:** 2
**Contribution:** 2
**Rating:** 2
**Confidence:** 5

**Summary:**

This work benchmarks how well LLMs understand and reason over CTI from different sources. The benchmark includes 9 tasks across structured, unstructured, and hybrid CTI settings, using 691 question–answer pairs. The authors evaluated 10 popular LLMs under both closed-book and knowledge-augmented settings.

**Strengths:**

1. The work covers structured, unstructured, and hybrid CTI tasks.

2. The tasks reflect what security analysts actually do (e.g., mapping CVEs to weaknesses, profiling threat actors).

**Weaknesses:**

1. The benchmark contains only 691 QA instances, which may not be enough to reflect the complex and ever-evolving nature of CTI.

2. Much of the dataset is produced through LLM prompting with human filtering rather than entirely human-authored. It is thus unclear  about biases introduced during such a semi-automation process.

3. Using fixed templates with predefined associations may lack the linguistic and structural diversity seen in real-world CTI queries. This may limit how well the benchmark tests genuine reasoning ability versus pattern-matching on fixed formats.

4. The benchmark assumes that entities like CVEs, malware, or threat actor names can be cleanly mapped across sources using templates and matching rules. In reality, CTI is full of conflicts and ambiguous or conflicting evidence, which the benchmark largely ignores rather than tackles directly.

**Questions:**

Please see the comments above.

---

> ### Author Response · Authors · 2025-11-20
> **Thank you for your valuable feedbacks.**
>
> ### Q: The benchmark contains only 691 QA instances, which may not sufficiently reflect CTI complexity.
>
> **A:** We appreciate the concern. Benchmark quality is determined not by raw volume but by whether the task scope is comprehensive and representative. In CTIArena, hybrid and unstructured tasks require multi-annotator verification by CTI practitioners, making them unscalable to significantly larger sizes.
>
> In response to reviewer feedback, we expanded the dataset during the rebuttal period to **1,433 instances**:
>
> - **303** unstructured
> - **641** structured
> - **489** hybrid (including two new tasks: **ECA** for NL→CAPEC and **ACA** for NL→CVE)
>
> We re-evaluated frontier closed-source models and lightweight open-source baselines; larger open-source models are still running due to resource constraints. **Model ranking and relative gaps on all existing task types remain fully consistent** with the original results, indicating that CTIArena’s conclusions are robust to increased scale.
>
> For the new hybrid tasks, the trends align with existing hybrid patterns:
>
> - **ECA:** GPT-5 achieves **0.784 (closed-book)** and **0.865 (RAG)**; other models cluster around 0.4–0.5.
> - **ACA:** Overall more difficult. GPT-5 reaches **0.401**, others mostly ≤0.28; **RAG improves GPT-5 to 0.692**.
>   - Difficulty stems from semantic mismatch: CVE entries follow templated phrasing, while CTI reports use scenario-driven descriptions with limited lexical overlap.
>
> The expanded dataset and updated results will be incorporated into the revised manuscript.
>
> ---
>
> ### Q: Much of the dataset is generated via LLM prompting with human filtering. Could this introduce bias?
>
> **A:** This setup does not introduce bias. We use LLM-assisted data construction solely to achieve scalable dataset generation; a fully manual pipeline would be prohibitively costly and would severely limit dataset size. To prevent bias, we adopt protocols that strictly separate LLM-generated surface text from the construction of ground-truth annotations.
>
> - **Structured tasks:** All mappings (e.g., CVE→CWE) come directly from authoritative CTI databases such as NVD and MITRE ATT&CK. These relations are official, deterministic, and never model-generated.
> - **Hybrid tasks:** Alignments between narrative report segments and CTI entities are derived through human reading and manual searching for corresponding CTI entries.
> - **Unstructured tasks:** Cross-report evidence synthesis is similarly manual: annotators read multiple CTI reports, identify evidence spans, and derive consolidated answers.
>
> All annotations undergo **three-annotator verification**: the first two annotators provide independent answers (inter-annotator agreement **κ = 0.73**), and a third annotator resolves conflicts.
> Because all annotations are authoritative or manually established, the LLM’s limited role in surface phrasing and filtering cannot introduce bias.
>
> ---
>
> ### Q: Fixed templates may reduce linguistic diversity and encourage pattern matching rather than reasoning.
>
> **A:** This setup does not reduce linguistic diversity or encourage pattern matching. Our prompt design is explicitly **non-fixed**:
>
> - For unstructured tasks, each question is generated using a general instruction plus a randomly sampled **intent slot** specifying the analytic focus (e.g., region, sector, tooling, remediation, detection).
> - For every sample, one or more intent slots are drawn from a large pool, ensuring high linguistic variability while maintaining semantic correctness.
> - The general instruction itself has multiple paraphrased variants, and the pipeline randomly selects among them.
>
> This combination of diversified instructions and randomized intent slots avoids uniform phrasing and ensures that the resulting questions require genuine CTI reasoning rather than pattern memorization.
>
> ---
>
> ### Q: The benchmark assumes clean one-to-one mappings, but real CTI often contains ambiguous or conflicting evidence.
>
> **A:** We agree that operational CTI includes ambiguity and conflicting reports. However, diagnostic benchmarks require **stable and reproducible ground truth**.
> CTIArena intentionally restricts to cases with clear and authoritative one-to-one mappings, removing instances where multiple interpretations are equally plausible. Examples of excluded samples include cases with:
>
> - multiple plausible ATT&CK techniques, or
> - insufficient evidence for CWE attribution.
>
> This avoids introducing subjective judgments and ensures that model differences reflect reasoning ability rather than disagreements over ground truth.
>
> CTIArena already represents a significant step for CTI benchmarking: it establishes principled task categories, integrates knowledge-augmented LLM evaluation, and demonstrates that even under **simplified and unambiguous** CTI settings, current LLMs still struggle.
> Uncertainty modeling is important, but it is a **next-stage direction** motivated by the challenges surfaced through CTIArena.

---

### Official Review · Reviewer_6ADC · 2025-11-03

**Soundness:** 3
**Presentation:** 3
**Contribution:** 2
**Rating:** 4
**Confidence:** 2

**Summary:**

(1) The paper introduces a benchmark for CTI reasoning. The benchmark is constructed through a 3-stage process:
- stage 1: annotate correlation among different sources
- stage 2: generating QA tasks
- stage 3: human-curator

(2) Empirical evaluation is performed by comparing several target models under different setups (closed-domain vs. knowledge-augmented), as shown in Table II.

(3) Based on these experiments, the paper presents several observations about CTI reasoning performance and challenges.

**Strengths:**

- The proposed benchmark is timely and relevant, contributing to standardized evaluation of cybersecurity-related agent.

- The proposed benchmark includes tasks that expand coverages of existing benchmarks.

**Weaknesses:**

- The paper employs GPT-5 as an LLM-based judge to assess data quality during dataset construction and as an automatic evaluator to score model predictions against reference answers in the testing phase. However, the authors also include GPT-5 in the performance comparison (Tables 2, 3, and 4), emphasizing its superiority. This practice may introduce evaluation bias, as the same model is used both as a judge and as a participant in the comparison.

- The evaluation process lacks detailed descriptions and does not include the evaluation of fine-tuned LLMs. In the Structured Tasks of Table 2, after injecting the related official entries of structured enumerations into the LLM, the performance improves significantly: from around 0 in the closed-book setting to around 1 with inference-time knowledge injection. There should be analysis to identify and verify the precise sources of the observed performance improvements.

- Interpretation of results is unclear. It is not entirely evident how to interpret Table II on the proposed  benchmark’s overall validity.

- Although Table I provides examples illustrating the benchmark’s coverage, there is no clear quantitative measure of how comprehensive or representative the benchmark is.

**Questions:**

- Identify the source of the significant  jump of improvement as mentioned earlier.

- A quantitative comparison with other benchmark.

---

> ### Author Response · Authors · 2025-11-20
> **Thank you for your valuable feedbacks.**
>
> ### Q: The paper uses GPT-5 both as an evaluator and as one of the evaluated models. Does this introduce bias?
>
> **A:** This setup does not introduce bias. Using strong LLMs as judges is a standard and widely adopted protocol in recent benchmarks (e.g., MT-Bench-101, Agent-as-a-Judge) to enable automatic and scalable evaluation.
> We further validated its reliability through a human-calibration study on 100 anonymized examples containing outputs from all evaluated models. Human preferences show high agreement with GPT-5’s judgments (**Cohen’s κ = 0.82**), indicating that GPT-5 does not systematically favor itself.
>
> During dataset construction, GPT-5 was used only for two constrained purposes:
> 1. **Generating surface-level natural-language QA pairs** from human-verified annotations and predefined templates.
> 2. **Filtering clearly low-quality instances** using a fixed rubric.
>
> All underlying annotations come exclusively from authoritative CTI sources or human validation; GPT-5 never introduces domain knowledge or labels. This separation ensures that its use does not bias the constructed dataset.
>
> ---
>
> ### Q: In the structured tasks, performance rises from near-zero in closed-book to almost perfect with knowledge injection. What explains this?
>
> **A:** The improvement reflects two known limitations of parametric LLM knowledge.
> First, many structured CTI mappings (e.g., CVE→CWE, CVE→CAPEC) fall into the long tail of an LLM’s parametric knowledge and are therefore difficult to recall reliably in closed-book settings.
> Second, CTI knowledge evolves quickly, and a non-trivial portion of these mappings was created or updated after model training cutoffs. When relevant facts are absent from parametric memory, models tend to guess or hallucinate.
>
> When authoritative entries from CTI databases are retrieved and provided as context, models can perform reliable in-context reasoning even when this context contradicts outdated parametric knowledge.
> The gains therefore demonstrate the **effectiveness and necessity of knowledge injection**.
>
> ---
>
> ### Q: Why are fine-tuned LLMs excluded?
>
> **A:** Fine-tuned CTI models typically encode a fixed knowledge snapshot and are optimized for one task family. They are therefore not suitable as baselines for CTIArena, which spans a broad range of task types and interacts with diverse data sources.
> Baselines in CTIArena instead **decouple LLM reasoning from the underlying knowledge**: models retrieve up-to-date intelligence and use reasoning to integrate the retrieved evidence.
>
> ---
>
> ### Q: How should Table II be interpreted? Does it support the benchmark’s validity?
>
> **A:** “KW” denotes the strongest **knowledge-augmented** configuration for each task type.
> The performance pattern shows a consistent **difficulty gradient** across models:
>
> - **Structured tasks:** Largest gains because retrieval is precise and the candidate space is small.
> - **Hybrid tasks:** Still challenging; grounding narrative text to abstract CTI enumerations introduces semantic variability, and retrieval errors propagate to predictions.
> - **Unstructured tasks:** Hardest; require multi-document retrieval and evidence synthesis across heterogeneous sources.
>
> This gradient reflects the **uneven landscape of CTI reasoning difficulty** for LLMs and highlights where current models struggle most.
>
> ---
>
> ### Q: Can CTIArena’s representativeness be quantified?
>
> **A:** Representativeness is quantified in the Related Work section through a task-family-level comparison with CTIBench and SEvenLLM. Because these benchmarks differ in task formulation, raw instance counts are not directly comparable; **coverage of CTI task families** is a more meaningful basis for comparison.
>
> During the rebuttal period, CTIArena was expanded to **1,433 samples**:
>
> - **303** unstructured
> - **641** structured
> - **489** hybrid
>   - including two new tasks: **ECA** (NL→CAPEC) and **ACA** (NL→CVE)
>
> We re-evaluated frontier closed-source models and lightweight open-source baselines; larger open-source models are still running due to resource constraints.
> **Model ranking and relative gaps remain consistent** with the original results, showing robustness under increased scale.
>
> For the new hybrid tasks:
>
> - **ECA:** GPT-5 achieves **0.784 (closed-book)** and **0.865 (RAG)**; others cluster around ~0.4–0.5.
> - **ACA:** Overall harder. GPT-5 reaches **0.401**, others mostly ≤0.28; **RAG improves GPT-5 to 0.692**.
>   - Difficulty stems from semantic mismatch: CVE entries follow templated phrasing, while CTI reports use scenario-driven descriptions with limited lexical overlap.
>
> These expanded results will be incorporated into the revised manuscript.

---

### Meta-Review · Area_Chair_Rf3f · 2026-01-04

**Summary:**

The paper presents CTIArena, a new benchmark designed to evaluate LLMs across heterogeneous Cyber Threat Intelligence sources. The benchmark covers nine tasks categorized into structured, unstructured, and hybrid formats. While the reviewers acknowledge the importance and timeliness of a standardized CTI benchmark, the submission faces significant concerns regarding the fundamental nature of the tasks, specifically, whether they evaluate "reasoning" or merely "factual lookup", and the potential for evaluation bias. Despite a proactive rebuttal where the authors doubled the dataset size and provided calibration studies, the core criticism regarding the "lookup" nature of the benchmark remains unresolved for the majority of the reviewers. For a "reasoning" benchmark, the tasks must require more than the successful retrieval of a mapping from a database. The paper is not yet ready for acceptance in its current form.

**Reviewer Concerns:**

Both Reviewers qfcD and oucB expressed concern that 691 QA instances were insufficient. The authors addressed this by more than doubling the dataset to 1,433 instances during the rebuttal period. Reviewer 6ADC worried that using GPT-5 to judge a competition it also participated in would introduce bias. The authors mitigated this with a human-calibration study of 100 samples, showing a high Cohen’s of 0.82, which suggests the model's judgments align with human experts and do not systematically favor itself. While the authors argue that "schema-level relational reasoning" is a distinct skill, Reviewer oucB maintains that if a task can be "trivially solved via a direct database or Google search," it does not test genuine reasoning ability. The nearly perfect performance jump after knowledge injection reinforces the reviewer's view that these are primarily retrieval tasks. Reviewer qfcD argued that using fixed templates might limit the benchmark to testing "pattern-matching on fixed formats" rather than real-world query understanding. While the authors explained their use of randomized "intent slots" to vary phrasing, the reviewer’s concern about whether the benchmark captures the "linguistic and structural diversity" of real-world CTI operations persists.

**Reviewer Scores:**

Reviewer oucB held a fundamental, philosophical disagreement with the paper’s definition of "reasoning". They argued that the tasks are "lookup problems" that LLMs are not the appropriate tool for. The authors' defense—that verifying relational constraints in a CTI schema is reasoning—directly contradicts this reviewer’s stated position that such tasks are "trivial". Furthermore, the reviewer accused the authors of overstating novelty compared to CTIBench. Even after the authors admitted to a "typo" regarding CTIBench's size, the core disagreement over what constitutes "reasoning" is so central to this reviewer's critique that a score change is unlikely.

---

### Decision · Program_Chairs · 2026-01-26

Reject